# Noise-conditioned Energy-based Annealed Rewards (NEAR): A Generative Framework for Imitation Learning from Observation

**Anish Abhijit Diwan**[1*]**, Julen Urain**[2,5]**, Jens Kober**[1†]**, Jan Peters**[2,3,4,5†]

[1]Department of Cognitive Robotics, TU Delft, Netherlands
[2]Department of Computer Science, TU Darmstadt, Germany
[3]Hessian Center for Artificial Intelligence (Hessian.ai), Germany
[4]Center for Cognitive Science, TU Darmstadt, Germany
[5]German Research Center for AI (DFKI)
[†]Equal supervision

## Abstract

This paper introduces a new imitation learning framework based on energy-based generative models capable of learning complex, physics-dependent, robot motion policies through state-only expert motion trajectories. Our algorithm, called Noise-conditioned Energy-based Annealed Rewards (NEAR), constructs several perturbed versions of the expert's motion data distribution and learns smooth, and well-defined representations of the data distribution's energy function using denoising score matching. We propose to use these learnt energy functions as reward functions to learn imitation policies via reinforcement learning. We also present a strategy to gradually switch between the learnt energy functions, ensuring that the learnt rewards are always well-defined in the manifold of policy-generated samples. We evaluate our algorithm on complex humanoid tasks such as locomotion and martial arts and compare it with state-only adversarial imitation learning algorithms like Adversarial Motion Priors (AMP). Our framework sidesteps the optimisation challenges of adversarial imitation learning techniques and produces results comparable to AMP in several quantitative metrics across multiple imitation settings. Code and videos available at anishhdiwan.github.io/noise-conditioned-energy-based-annealed-rewards/

## 1 Introduction

Learning skills through imitation is probably the most cardinal form of learning for human beings. Whether it is a child learning to tie their shoelaces, a dancer learning a new pose, or a gymnast learning a fast and complex manoeuvre, acquiring new motor skills for humans typically involves guidance from another skilled human in the form of demonstrations. Acquiring skills from these demonstrations typically boils down to interpreting the individual features of the demonstration motion – for example, the relative positions of the limbs in a dance pose – and subsequently attempting to recreate the same features via repeated trial and error. Imitation learning (IL) is an algorithmic interpretation of this simple strategy of learning skills by matching the features of one's own motions with the features of the expert's demonstrations.

Such a problem can be solved by various means, with techniques like *behavioural cloning (BC)*, *inverse reinforcement learning (IRL)*, and their variants being popular choices (Osa et al., 2018). The imitation learning problem can also be formulated in various subtly differing ways, leading to different constraints on the types of algorithms that solve the problem. One notably challenging version of the problem is *Imitation from Observation* (IfO) (Torabi et al., 2018; 2019; Zare et al., 2024), where the expert trajectories are only comprised of state features and no information about the expert's actions is available to the imitator. This means that learning a policy is not as straightforward as capturing the distribution of the expert's state-action pairs. Instead, the imitator must also

---

[*]Corresponding author: Anish Abhijit Diwan (`anishhdiwan@gmail.com`)

learn to capture the dynamics of its environment. From a practical perspective, the IfO problem is quite relevant as obtaining action-rich data for real-world tasks – across several agent embodiments and at large scales – is rather challenging. In most tasks, the expert only has an implicit representation of the policy. Imagine how a dancer cannot realistically convey their low-level actions – like muscle activations or positional targets – in a dance routine. Further, collecting action-rich data by teleoperating the agent requires significant human effort and often offers limited motion complexity. Imitation from observation hence closely depicts the data-limited reality of applying IL in the real world. Unfortunately, because BC relies on the expert's actions, a large fraction of BC techniques (including state-of-the-art diffusion-based algorithms like (Chi et al., 2023)) are inapplicable to the problem of imitating from observation. Inverse reinforcement learning, on the other hand, can still be applied to such problems.

In this work, we mainly focus on observation-based inverse reinforcement learning, where the agent recovers a scalar reward signal from the demonstrations that when maximised by updating the agent's policy, provides the agent with the "correct" motivation to imitate the expert. While reward learning in itself is a broad field of study, recent works that leverage generative adversarial techniques for this task have shown markedly good results (Tessler et al., 2023; Peng et al., 2021; Ho & Ermon, 2016; Torabi et al., 2018). The fundamental idea in adversarial imitation learning (AIL) is to simultaneously learn and optimise the return from the reward function implied in the expert demonstrations through an optimisation objective derived from *Generative Adversarial Networks* (GANs) (Goodfellow et al., 2014). This GAN-inspired min-max optimisation procedure considers the agent's closed-loop policy as a generator and simultaneously trains a discriminator to differentiate between the motions in the expert data distribution and the motions produced by the agent's policy. The discriminator aims to correctly label samples from both distributions while the generator aims to return an action that when applied to the environment, leads to features that resemble those in the expert's data distribution. The discriminator's prediction is used as a reward signal in reinforcement learning (RL) and the policy and the discriminator are updated iteratively until convergence. Although methods like AMP (Peng et al., 2021), GAIL (Ho & Ermon, 2016), GAIfO (Torabi et al., 2018) (purely adversarial IL), and DiffAIL (Wang et al., 2024), DIFO (Huang et al., 2024) (diffusion enhanced adversarial IL) have achieved impressive results in a wide variety of imitation tasks, they are prone to challenges intrinsic to their theoretical formulation. The simultaneous min-max optimisation used to learn the reward function in these techniques is highly sensitive to hyperparameter values. This causes adversarial learning techniques to have unstable training dynamics and learn non-smooth probability densities (Saxena & Cao, 2021; Kodali et al., 2017; Arjovsky & Bottou, 2017; Goodfellow et al., 2014). These limitations ultimately lead to instability and poor reinforcement learning when using adversarial techniques to learn reward functions.

This paper explores a non-adversarial generative framework for reward learning that completely sidesteps the limitations of previous generative imitation learning techniques. Our primary contribution is to use energy-based generative models as the backbone of the reward learning framework to learn smooth and accurate representations of the data distribution. We propose to use the learnt energy functions as reward functions and present a new imitation learning algorithm called *Noise-conditioned Energy-based Annealed Rewards (NEAR)* that has better, **more stable learning dynamics, and learns smooth and unambiguous reward signals.** NEAR produces motions comparable to state-of-the-art adversarial imitation learning methods like AMP Peng et al. (2021). Before diving into our proposed framework (Sections 4 and 5), we first briefly discuss the challenges of adversarial IL in Section 2 and score-based generative modelling in Section 3.

## 2    BACKGROUND: ADVERSARIAL IMITATION LEARNING

Given an expert motion dataset $\mathcal{M}$ containing i.i.d. data samples $x \equiv (s, s') \in X$ implying a distribution $p_D$, where $X$ is the space of state transitions [1], adversarial IL methods aim to learn a differentiable generator (policy) $\pi_{\theta_G}(s) : S \to A$ where $S$ is the state space, $A$ is the action space, and $s \in S$ is a sample drawn from the occupancy measure of the policy $\rho_\pi$. Similarly to a standard GAN, the idea here is to learn a differentiable discriminator $D_{\theta_D}(x) : X \to [0, 1]$ that returns a scalar value representing the probability that the sample $x$ was derived from $p_D$. However,

---

[1]In this paper we define all expressions for the partially observable case, however, the same results also apply to the fully observable cases.

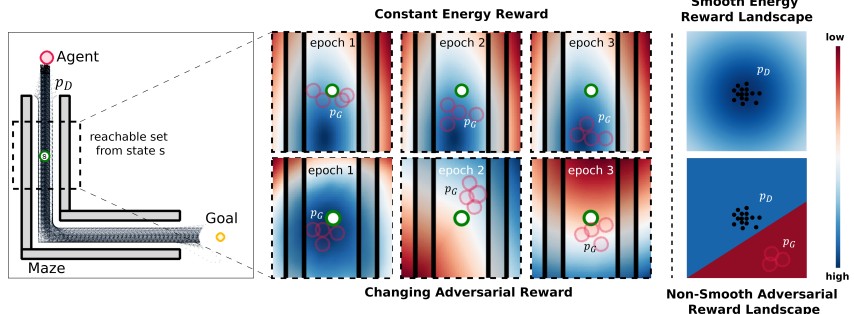

Figure 1: A comparison of reward functions (probability density approximations) learnt in a 2D target-reaching imitation task (left). In this task, an agent aims to reach a goal and expert demonstrations ($p_D$) pass through an L-shaped maze. The learnt reward function is expected to encourage the agent to pass through the maze. In the middle, we show $\texttt{rew}(s'|s)$ for all reachable states around a state $s$ (green circle) at different training epochs. On the right, we show an illustration of the non-smooth reward landscape of adversarial IL. The energy-based reward is a smooth (with continuous gradients), accurate representation of $p_D$ and is constant regardless of the distribution of policy-generated motions ($p_G$). In contrast, the adversarial reward is non-smooth and prone to instability. Additionally, it changes depending on $p_G$ (the discriminator tends to minimise policy predictions) and can provide non-stationary reward signals (additional details in Appendix B.3).

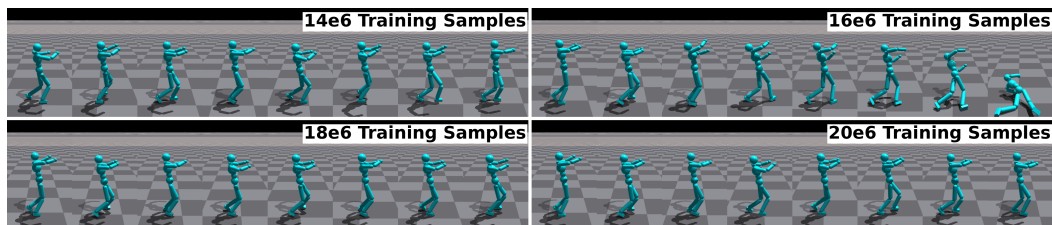

Figure 2: Degradation of an adversarially learnt policy (AMP) in a stylised walking imitation task. With sufficient training, the policy does learn to complete the task, however, performance fluctuates substantially throughout training (with degradation seen at $16e6$ training samples).

now there exists an additional function $W(\pi_{\theta_G}(s)) : A \to X$ that maps the output of the policy to the discriminator's input space. The discriminator is learnt assuming that $W(\pi_{\theta_G}(s))$ is an i.i.d. sample in $X$ and the generator is learnt via policy gradient methods by using $\log D_{\theta_D}(W(\pi_{\theta_G}(s)))$ as a reward function (Peng et al., 2021; Torabi et al., 2018). The AIL optimisation procedure is as follows ($J$ is the performance measure as per the policy gradient theorem Sutton et al. (1999)).

$$\min_{\theta_D} \mathbb{E}_{x \sim p_D}[\log D_{\theta_D}(x)] + \mathbb{E}_{s \sim \rho_{\pi_{\theta_G}}}[\log(1 - D_{\theta_D}(W(\pi_{\theta_G}(s))))] \tag{1}$$

$$\max_{\theta_G} J \text{ where } \nabla_{\theta_G} J(\pi_{\theta_G}) = \mathbb{E}_{\pi_{\theta_G}}[Q^{\pi_{\theta_G}}(s,a)\nabla_{\theta_G} \log \pi_{\theta_G}(s,a)]$$

$$\text{where } Q^{\pi_{\theta_G}}(s,a) = \mathbb{E}_{\pi_{\theta_G}}[\log D_{\theta_D}(W(\pi_{\theta_G}(s)))]$$

Similarly to standard sample-generation-focused GANs, AIL algorithms also suffer from having a perfect discriminator (Arjovsky & Bottou, 2017). This leads to low discriminator predictions on the policy-generated samples, causing zero or constant rewards for the RL policy and ultimately leading to poor training dynamics. Moreover, the iterative nature of the AIL problem leads to a constantly changing manifold of policy-generated samples and an arbitrarily changing discriminator decision boundary. This causes unpredictability in the rewards and introduces drastic non-stationarity in the RL problem. Lastly, the rewards learnt via adversarial techniques are non-smooth and do not always provide an unambiguous signal for improvement. Here, smoothness refers to the ability of a reward

function to convey informative gradients in the sample space [2]. These challenges compound to cause fluctuations in the learnt policy and convergence to local minima (elaborated in Appendix E).

To conclude this section, we point to Figures 1 and 2 that show a visual example of the non-smooth nature of the discriminator and qualitative results demonstrating instability of an adversarially learnt policy. Having discussed several issues with adversarial IL, the next section introduces an alternative method of learning the expert's data distribution using which we subsequently propose a new imitation learning algorithm.

## 3    NOISE-CONDITIONED SCORE NETWORKS (NCSN)

Score-based generative models are a family of techniques recently popularised for generating realistic images and video samples. They model the unknown data distribution as a Boltzmann distribution and generate samples by an iterative denoising process by traversing along the gradient of the data distribution's log probability (Song & Kingma, 2021). Score-based models approximate the gradient of the log probability (called the score function) through a procedure called denoising score matching (Vincent, 2011). Similarly to GANs, the aim here is to learn a probability distribution $p_G$ that closely resembles the data distribution $p_D$, with $p_G \triangleq e^{-E(x)}/Z$ (Boltzmann distribution) and $E(x)$ called the energy function of the distribution. Intuitively, the energy function is a measure of the closeness of a sample to $p_D$ while the score ($\nabla_x \log p_G(x)$) is a vector pointing towards the steepest increase in the likelihood of $p_D$. Learning the score function implicitly also learns the energy function as $\nabla_x \log p_G(x) = \nabla_x \log(e^{-E(x)}/Z) = -\nabla_x E(x)$. In this paper, we propose to explicitly learn the energy function to then use the energy of a sample to guide reinforcement learning. To do so, we make modifications to a score-based framework called Noise Conditioned Score Networks (NCSN) (Song & Ermon, 2019; 2020).

The underlying idea of NCSN is to learn the score function by a process of artificial perturbation and denoising. Data samples are first perturbed by adding variable amounts of Gaussian noise. The score function is then learnt by estimating the denoising vectors that point from the perturbed data samples to the original ones. Given i.i.d. data samples $\{x \sim p_D \in \mathbb{R}^D\}$, NCSN (Song & Ermon, 2019) formulates a perturbation process that adds Gaussian noise $\mathcal{N}(x, \sigma)$ to each sample $x$, where $\sigma$ is the standard deviation representing a diagonal covariance matrix and is sampled uniformly from a geometric sequence $\{\sigma_1, \sigma_2, ..., \sigma_L\}$. Following this perturbation process, we obtain a conditional distribution $q_\sigma(x'|x) = \mathcal{N}(x'|x, \sigma I)$ from which a marginal distribution $q_\sigma(x')$ can be obtained as $\int q_\sigma(x'|x)p_D(x)dx$. Given this perturbed marginal distribution, NCSN attempts to learn a score function $s_\theta(x, \sigma) : \mathbb{R}^D \to \mathbb{R}^D$ that points from the perturbed samples back to the original ones. The idea is to learn a conditional function to jointly estimate the scores of all perturbed data distributions, i.e., $\forall \sigma \in \{\sigma_i\}_{i=1}^L : s_\theta(x', \sigma) \approx \nabla_{x'} \log q_\sigma(x')$. The score network is learnt via denoising score matching (DSM) (Vincent, 2011) on samples drawn from the conditional distribution $q_\sigma(x'|x)$ [3]. The final DSM loss is averaged over the various $\sigma$ values assigned to data samples in the training batch.

By perturbing individual data samples with Gaussian noise, NCSN essentially creates a perturbed distribution that is a smooth and dilated version of $p_D$ (illustrated in Figure 1) – with the standard deviation $\sigma$ controlling the level of dilation. This perturbation strategy ensures that $p_D$ is supported in the whole sample space and not just a low-dimensional manifold in $\mathbb{R}^D$ (manifold hypothesis (Fefferman et al., 2016; Cayton et al., 2008)), ensuring well-defined gradients and allowing a better score function approximation. It also ensures that the score function is accurately approximated in data-sparse regions in $p_D$ by increasing sample density in such regions (Song & Ermon, 2019).

## 4    NOISE-CONDITIONED ENERGY-BASED ANNEALED REWARDS (NEAR)

In NCSN, the perturbed conditional distribution $q(x'|x)$ is formulated as a Boltzmann distribution such that $q(x'|x) = e^{\text{DIST}(x',x)}/Z$ where $\text{DIST}()$ is a function that defines some distance measure

---

[2]While the gradient of the reward function is not a direct part of the policy update in policy gradient methods, a smooth reward function is necessary for "sensible" policy updates.

[3]We leverage the fact that $\mathcal{L}_{\text{DSM}}(q_\sigma(x')) = \mathcal{L}_{\text{DSM}}(q_\sigma(x'|x)) + \text{const.}$ (Song & Kingma, 2021; Vincent, 2011).

---

**Algorithm 1:** Noise-conditioned Energy-based Annealed Rewards

---

**Data:** $\mathcal{M} \equiv \{(s, s')\}, \{\sigma_i\}_{i=1}^L$

Initialise energy network $e_\theta$, policy $\pi_{\theta_G}$, and rollout horizon ;        `// Appendix B.2.2`

Initialise replay buffer $\mathcal{B} \leftarrow \emptyset$, annealing buffer $\mathcal{A} \leftarrow \emptyset$, and annealing threshold $\alpha$

**Subroutine** `EnergyNCSN()`:

    **while** *not done* **do**

        $b^{\mathcal{M}} \leftarrow$ sample a batch of transitions from $\mathcal{M}$ ;

        $b^{sigma} \leftarrow$ sample a batch of noise levels $\sigma_k$ uniformly from $\{\sigma_i\}_{i=1}^L$ ;

        Update $e_\theta$ according to Equation (2) using pairing $b^{\mathcal{M}} : b^{sigma}$

    **end**

**Subroutine** `RL()`:

    Initialise $\sigma_k = \sigma_1$

    **while** *not done* **do**

        **for** $i = 0, 1, \cdots$ **do**

            $\tau_i \leftarrow \{[s, a, s', r' = \text{rew-tf}(e_\theta(s, s', \sigma_k))]^{\text{till horizon}}\}_{\pi_{\theta_G}}$ ;     `// Section 5.1`

            Store $\tau_i$ in $\mathcal{B}$ & $\{(s, s')^{\text{till horizon}}\}$ in $\mathcal{A}$

        **end**

        progress $= \frac{e_\theta(\mathcal{A}, \sigma_k)}{\text{mean energy on switching to } \sigma_k} - 1$ ;

        Switch $\sigma_k = \sigma_{k+1}$ if progress $> \alpha$ and $\sigma_k = \sigma_{k-1}$ if progress $< -\alpha$ ; `// Annealing`

        $\mathcal{A} \leftarrow \emptyset$ & update $\pi_{\theta_G}$ using $\mathcal{B}$

    **end**

**Algorithm:**

    Call `EnergyNCSN`

    Call `RL`

---

between a sample and its perturbed form. In our case, $\text{DIST}_\sigma()$ is the energy function of a Gaussian distribution with $\sigma$ standard deviation and is a smooth, dilated representation of the energy landscape of the expert data distribution $p_D$. Our approach leverages the fact that for a sample $x \in X$, $\text{DIST}_\sigma(x)$ is essentially just a scalar-valued measure of the closeness of $x$ to $p_D$, meaning that it can be used as a reward signal to guide a policy to generate motions that gradually resemble those in $p_D$.

This approach sidesteps several of the shortcomings of adversarially learnt rewards. Since $\text{DIST}_\sigma()$ is learnt via score matching on samples arbitrarily far away from the $p_D$, it is both well-defined and continuous in the relevant parts of the sample space $X$. Further, since $\text{DIST}_\sigma()$ is a dilated version of the energy function of $p_D$, it is also not prone to being constant valued. Continuity and informativeness in the whole space indeed require an infinitely large $\sigma$, however, realistically a $\sigma$ that is sufficiently large to cover the worst-case policy would guarantee that the rewards are both smooth and non-constant in the parts of $X$ where the policy-generated samples are realistically expected to lie (proof: Appendix A.1). Moreover, $\text{DIST}_\sigma()$ is learnt using perturbed samples from $p_D$ and hence does not rely on the policy-generated samples. This means that it is disconnected from the policy and is not prone to issues of high variance that come with simultaneous training. Finally, we propose to train NCSN *before* training the reinforcement learning policy, eliminating any concerns of non-stationarity. The following sections discuss the procedure to learn these energy functions and other algorithmic details of our approach (Algorithm 1).

## 4.1 LEARNING ENERGY FUNCTIONS

Given $D$-dimensional i.i.d. data samples $\{x \sim p_D \in \mathbb{R}^D\}$ where $p_D$ is the distribution of state-transition features in the expert's trajectories, NEAR learns a parameterised energy function $e_\theta(x', \sigma) : \mathbb{R}^D \rightarrow \mathbb{R}$ that approximates the energy of samples $x'$ in a perturbed data distribution obtained by the local addition of Gaussian noise $\mathcal{N}(x, \sigma)$ to each sample $x$. The idea here is to jointly estimate the energy functions of several perturbed distributions, i.e., $\forall \sigma \in \{\sigma_i\}_{i=1}^L : e_\theta(x', \sigma) \approx \text{DIST}_\sigma(x')$. The sample's score is computed by taking the gradient of the predicted energy w.r.t. the perturbed sample, $s(x', \sigma) = \nabla_{x'} e_\theta(x', \sigma)$. The energy network is learnt via denoising score matching (DSM) (Vincent, 2011) using this computed

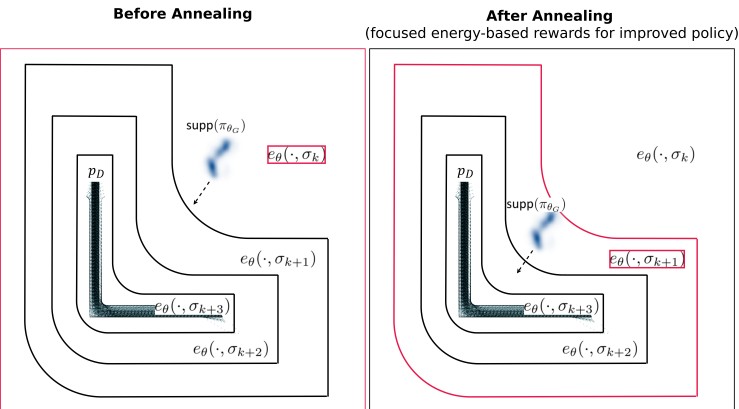

Figure 3: Annealing (during RL) ensures that the agent always receives a focused and well-defined reward signal, thereby motivating the policy to produce motions similar to $p_D$. Here, $p_D$ is a distribution of the expert's state transitions in a 2D target-reaching task (introduced in Figure 1). The learnt energy functions $e_\theta(\cdot, \sigma_k)$ are illustrated as dilated versions (L-shaped boundaries) of $p_D$ that are well-defined only inside their respective perturbed manifold ("inner" regions of the L-shaped boundaries). The manifold of policy-generated motions is indicated by $\text{supp}(\pi_{\theta_G})$. The policy is shown to have improved from left to right since $\text{supp}(\pi_{\theta_G})$ for the improved policy is closer to $p_D$. The reward function currently maximised by the agent is highlighted in red. During RL, the agent starts at the energy function (reward) of a lower noise level ($e_\theta(\cdot, \sigma_k)$) and switches to a higher one ($e_\theta(\cdot, \sigma_{k+1})$) upon receiving a sufficiently high average return. Arrows indicate the gradient of the rewards in $\text{supp}(\pi_{\theta_G})$ (avg. score).

score and the final DSM loss in a training batch is computed as an average over the various $\sigma$ values assigned to data samples in the training batch.

$$l_{\text{DSM}}(\sigma) = \frac{1}{2} \mathbb{E}_{p_D} \mathbb{E}_{x' \sim \mathcal{N}(x, \sigma)} \left[ \left\| \frac{x' - x}{\sigma^2} - \nabla_{x'} e_\theta(x', \sigma) \right\| \right]$$

$$\mathcal{L}_{\text{DSM}}(\{\sigma_i\}_{i=i}^L) \triangleq \frac{1}{L} \sum_{i=1}^L l_{\text{DSM}}(\sigma_i) \tag{2}$$

## 4.2 ANNEALING

We modify the definition of the perturbed conditional distribution $q(x'|x)$ by flipping the sign of the energy function such that higher energies indicate closeness to $p_D$. This is done to simplify the downstream reinforcement learning such that the predicted energy can be maximised directly. Following the improvements introduced in Song & Ermon (2020), we define $e_\theta(x', \sigma) = e_\theta(x')/\sigma$ where $e_\theta(x')$ is an unconditional energy network [4]. This allows us to learn the energy function of a large number of noise scales with a very small sized dataset.

The appropriate selection of the noise scale ($\{\sigma_i\}_{i=i}^L$) is highly important for the success of this framework. $\sigma_L$ must be small enough that the perturbed distribution $q_{\sigma_L}()$ is nearly identical to $p_D$. This ensures that the policy aims to truly generate samples that resemble those in $p_D$. In contrast, $\sigma_1$ must be sufficiently large such that $e_\theta(x', \sigma_1)$ is well-defined, continuous, and non-zero for any sample that is generated by the worst-possible policy. This ensures that the agent always receives an informative signal for improvement. Assuming that policy degradation is unlikely, $\sigma_1$ must be such that $\text{supp}(q_{\sigma_1}())$ effectively contains the support of the distribution induced by a randomly initialised policy network. In practice, these are dataset-dependent hyperparameters.

---

[4]The score is the gradient of the energy function and the norm of the score scales inversely with $\sigma$. The score can hence be approximated by rescaling the energy with $\frac{1}{\sigma}$ (Song & Ermon, 2020).

The trained energy network $e_\theta(x', \sigma)$ can directly be used as a reward function to train a policy network $\pi_{\theta_G}$ (say initialised at $\theta_{G0}$) using some fixed noise level $\sigma_k$. But how do we decide on an appropriate $\sigma_k$? Assuming that during training the noise scale was set appropriately, $q_{\sigma_L}()$ is nearly identical to $p_D$ but $\text{supp}(q_{\sigma_L}())$ has a low intersection with $\text{supp}(\pi_{\theta_{G0}})$ [5]. On the other hand $q_{\sigma_1}()$ is an extremely dilated version of $p_D$ but $\text{supp}(q_{\sigma_1}())$ is likely to have a high intersection with $\text{supp}(\pi_{\theta_{G0}})$. This means that any chosen noise-level $\sigma_k$ offers a tradeoff between sample quality and $e_\theta(x', \sigma_k)$ being continuous and well-defined in the manifold of the samples generated by the current policy.

We propose an annealing framework inspired by annealed Langevin dynamics and its predecessors (Song & Ermon, 2019; Kirkpatrick et al., 1983; Neal, 2001) to ensure that the learnt reward function is always well-defined and continuous while also gradually changing to motivate the policy to get closer to $p_D$ (illustrated in Figure 3). Instead of focusing on sample generation, annealing in the context of reinforcement learning focuses on making gradual changes to the agent's reward function. Our annealing framework hence depends on the agent's progress from an imitation perspective. Training is initialised with the energy function of the lowest noise level in the geometric noise scale. Then, at every new noise level in the scale, the agent tracks the average return of the first few policy updates. The noise level of the energy function is increased if the average return of the last few policy updates is higher than some percentage of the initial return. We note that changing the reward function introduces non-stationarity in the reinforcement learning problem, meaning that the learnt policy is susceptible to degradation. To account for this, our framework also lowers the noise level if the return drops below some percentage of the initial return. This means that if the policy gets worse, the noise level decreases, thereby increasing the intersection between $\text{supp}(q_\sigma())$ and $\text{supp}(\pi_{\theta_G})$ and ensuring that the degraded policy still has an informative reward signal for improvement.

## 5 EXPERIMENTS

### 5.1 EXPERIMENTAL SETUP

We evaluate NEAR (Algorithm 1) on complex, physics-dependent, contact-rich humanoid motions such as stylised walking, running, and martial arts. The chosen task set demands an understanding of physical quantities such as gravity and the mass/moments of inertia of the character and contains a variety of fast, periodic, and high-acceleration motions. The expert's motions are obtained from the CMU and SFU motion capture datasets and contain trajectories of several motions. For each motion, a dataset of state transitions $\mathcal{M} \equiv \{(s, s')\}$ is created to learn an imitation policy. Rewarding the agent for producing similar state transitions as the expert, incentivises the agent to also replicate the expert's unknown actions.

$$\tilde{r}(s, a, s', g) = w^{task} r^{task}(s, a, g) + w^{energy} e_\theta(s, s') \tag{3}$$

To understand the impact of motion data availability on the algorithm, we also train NEAR in a single-clip setting – using a single expert motion for training – on challenging motions like mummy-style walking and spin-kick. Further, to understand the composability of the learnt rewards, we train NEAR with both environment-supplied rewards (such as a target reaching reward) and energy-based rewards learnt from different motion styles (to perform hybrid stylised motions). To incorporate the environment-supplied task reward $r^{task}(s, a, g) \in [0, 1]$, we use the same strategy from Peng et al. (2021) and formulate learning as a goal-conditioned reinforcement learning problem, where the policy is now conditioned on a goal $g$ and maximises a reward $\tilde{r}(s, a, s', g)$ (Equation (3)). Details of the tasks and goals can be found in Appendix B.1. We also apply an additional reward transformation of $\tanh\left((\tilde{r} - r')/10\right)$ where $r'$ is the mean horizon-normalised return received by the agent in the last $k = 3$ policy iterations, $r' = \text{mean}(\{\bar{R}_{t-i}/horizon\}_{i=1}^{k})$. This bounds the unnormalised energy reward to a fixed interval so that changes between the noise levels $\sigma$ are smoother. Additionally, it grounds the agent's current progress in relation to its average progress in the last few iterations. The policy is trained using Proximal Policy Optimisation (Schulman et al., 2017) and we use the following quantitative metrics to measure the performance of our algorithm.

---

[5]As a shorthand, we abbreviate the support of the distribution of state transitions induced by rolling out a policy as $\text{supp}(\pi_{\theta_G})$.

Table 1: A comparison of the avg. pose error (shaded, lower is better) and spectral arc length (non-shaded, closer to expert is better) at the end of training. Stdev. across independent runs is shown as an error value (±).

| Algorithm | Walking (74 clips) | | Running (26 clips) | | Left Punch (19 clips) | | Crane Pose (3 clips) | | Mummy Walk (1 clip) | | Spin Kick (1 clip) | |
|---|---|---|---|---|---|---|---|---|---|---|---|---|
| NEAR | **0.51** ± 0.15 | **-7.52** ± 1.32 | **0.62** ± 0.17 | **-7.24** ± 1.59 | 0.37 ± 0.05 | **-6.87** ± 1.47 | 0.94 ± 0.15 | -6.6 ± 1.97 | 0.66 ± 0.39 | **-4.72** ± 1.2 | 0.78 ± 0.05 | -5.59 ± 2.26 |
| AMP | **0.51** ± 0.07 | -8.78 ± 1.04 | 0.65 ± 0.01 | -9.71 ± 1.54 | **0.32** ± 0.01 | -9.93 ± 3.28 | **0.82** ± 0.09 | **-8.1** ± 1.18 | **0.41** ± 0.01 | -13.84 ± 1.12 | **0.58** ± 0.1 | **-3.16** ± 0.73 |
| Expert | - | -5.4 | - | -3.79 | - | -1.73 | - | -12.28 | - | -4.71 | - | -3.39 |

Table 2: A comparison of the avg. pose error (shaded, lower is better) and task return (non-shaded, higher is better) at the end of training with composed reward functions. Stdev. across independent runs is shown as an error value (±).

| Algorithm | Target Reaching (walking) | | Target Reaching (running) | | Target Reaching & Punching | |
|---|---|---|---|---|---|---|
| NEAR | **0.94** ± 0.11 | **2.75** ± 0.82 | **1.18** ± 0.04 | **1.74** ± 0.46 | - | 3.6 ± 2.64 |
| AMP | 1.09 ± 0.13 | 2.23 ± 0.24 | 1.77 ± 0.31 | -0.15 ± 1.06 | - | **3.85** ± 0.76 |

***Average Dynamic Time Warping Pose Error:*** This is the mean dynamic time warping (DTW) error (Sakoe & Chiba, 1978) between trajectories of the agent's and the expert's poses averaged across all expert motions in the dataset. The DTW error is computed using $\|\hat{x}_t - x_t\|_2$ as the cost function, where $\hat{x}_t$ and $x_t$ are the Cartesian positions of the reference character and agent's bodies at time step $t$. To ensure that the pose error is only in terms of the character's local pose and not its global position in the world, we transform each Cartesian position to be relative to the character's root body position at that timestep ($\hat{x}_t \leftarrow \hat{x}_t - \hat{x}_t^{root}$ and $x_t \leftarrow x_t - x_t^{root}$).

***Spectral Arc Length:*** Spectral Arc Length (SAL) (Beck et al., 2018; Balasubramanian et al., 2011; 2015) is a measure of the smoothness of a trajectory and is an interesting metric to determine the policy's ability to perform periodic motions in a controlled manner. SAL relies on the assumption that smoother motions are comprised of fewer and low-valued frequency domain components while jerkier motions have a more complex frequency domain signature. It is computed by adding up the lengths of discrete segments (arcs) of the normalised frequency-domain map of a motion (in this case, we do not transform the positions to the agent's local coordinate system).

## 5.2 RESULTS

We compare Noise-conditioned Energy-based Annealed Rewards (NEAR) with Adversarial Motion Priors (AMP) (Peng et al., 2021) and in both cases only use the learnt rewards to train the policy. AMP is used as a baseline since it is an improved formulation of previous state-of-the-art techniques (Torabi et al., 2018; Ho & Ermon, 2016) and has shown superior results in the state-only adversarial IL literature. Each algorithm-task combination is trained 5 times independently and the mean performance metrics across 20 episodes of each run are compared. Both algorithms are trained for a fixed number of maximum iterations.

Figure 4 shows snapshots of the policies trained using NEAR. We find that NEAR achieves very close imitation performance with the expert's trajectory and learns policies that are visually smoother and more natural. For the quantitative metrics, we use the average performance at the end of training for a fair comparison and find that both NEAR and AMP are roughly similar across all metrics (Table 1). In most experiments, NEAR is closer to the expert in terms of the spectral arc length while AMP has a better pose error. NEAR also outperforms AMP in stylised goal-conditioned tasks, producing motions that both imitate the expert's style while simultaneously achieving the desired global goal (Table 2 and Figure 4 bottom). From the experiments on spatially composed learnt rewards, we find that NEAR can also learn hybrid policies such as waking while waving. Finally, we notice that NEAR performs poorly in single-clip imitation tasks, highlighting the challenges of accurately capturing the expert's data distribution in data-limited conditions. Conversely, AMP is less affected by data unavailability since the discriminator in AMP is simply a classifier and does not explicitly capture the expert's distribution.

## 5.3 ABLATIONS

We also conduct ablation experiments (Table 3) that help identify the crucial components of NEAR. The main focus of these experiments is to understand the contributions of annealing and the effects of

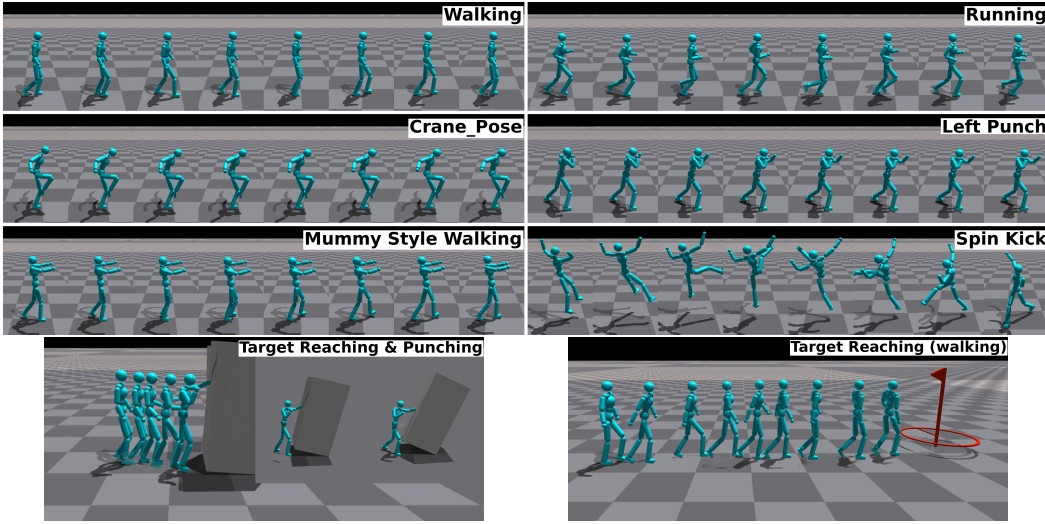

Figure 4: Snapshots of the policies trained with NEAR. Mummy-style walking and spin-kick are single-clip imitation tasks. The bottom row shows goal-conditioned RL policies that also optimise an environment-provided task reward.

Table 3: Avg. pose error (shaded, lower is better) and spectral arc length (non-shaded, closer to expert is better) in several ablated configurations of NEAR. Stdev. across independent runs is shown as an error value ($\pm$).

Effect of task rewards

| Config | Walking | | Running | | Crane Pose | |
|---|---|---|---|---|---|---|
| $\sigma_5$ & $e_\theta$ | $0.49 \pm 0.22$ | $-7.1 \pm 1.94$ | $0.62 \pm 0.18$ | $-6.69 \pm 1.85$ | $1.38 \pm 0.8$ | $-6.03 \pm 1.79$ |
| $\sigma_5$ & $\tilde{r}$ | $0.42 \pm 0.02$ | $-8.7 \pm 1.2$ | $0.57 \pm 0.04$ | $-8.61 \pm 0.86$ | $1.23 \pm 0.07$ | $-4.34 \pm 1.02$ |
| Expert | - | $-5.4$ | - | $-3.79$ | - | $-12.28$ |

Effect of annealing

| Config | Walking | | Running | | Crane Pose | |
|---|---|---|---|---|---|---|
| anneal & $e_\theta$ | $0.51 \pm 0.15$ | $-7.52 \pm 1.32$ | $0.62 \pm 0.17$ | $-7.24 \pm 1.59$ | $0.94 \pm 0.15$ | $-6.6 \pm 1.97$ |
| $\sigma_5$ & $e_\theta$ | $0.49 \pm 0.22$ | $-7.1 \pm 1.94$ | $0.62 \pm 0.18$ | $-6.69 \pm 1.85$ | $1.38 \pm 0.8$ | $-6.03 \pm 1.79$ |
| Expert | - | $-5.4$ | - | $-3.79$ | - | $-12.28$ |

using an environment-provided task reward (without goal-conditioning). For walking and running, the task reward favoured forward motion and episode length while for the crane pose task it only favoured episode length. Given these parameters of interest, we train ablated configurations of NEAR with either annealing or a reward function conditioned on a fixed noise level ($\sigma_5 \approx 9.21$) and with either only a learnt reward ($e_\theta$) or a composed reward function ($\tilde{r}$: Equation (3)).

It can be seen that the addition of the task reward leads to an improvement in the pose error. This is especially apparent in tasks like walking and running where the task reward is closely aligned with the imitation objective. It must however be noted that the addition of the task reward does mean that the agent has a reduced closeness to specific characteristics of the expert's motion like spectral arc length, velocity, and jerk. Ultimately the closeness of the imitation under a combined reward still highly depends on the harmony between the two reward functions.

Annealing does not have a significant impact on performance in walking and running, however, leads to an improvement in more complex, non-periodic cases like the crane-pose task. It is possible that for complex tasks, the expert distribution is more densely concentrated. In this case, a higher noise level for a complex task might be more informative than one for a simpler task for which the expert distribution is more spread out (examples in Appendix A.2). The increased information available from annealing might hence be the reason for better results with annealing in complex tasks.

## 6  LIMITATIONS & CONCLUSIONS

While NEAR is capable of generating high-quality, life-like motions and also outperforms AMP in several tasks, it is still prone to some limitations. The annealing strategy discussed in Section 4.2 does lead to progressively improving rewards, however, annealing at high noise levels also tends to cause unpredictability in the received rewards. We attribute this unpredictability to the static

nature of our annealing strategy. While the geometric nature of the noise scale indeed maximises the intersection between $\mathrm{supp}(q_{\sigma_k}())$ and $\mathrm{supp}(q_{\sigma_{k+1}}())$ (Song & Ermon, 2019), at higher values of $k$, a fixed percentage increase in the return does not always ensure that $\mathrm{supp}(\pi_\theta)$ is outside $\mathrm{supp}(q_{\sigma_k}()) \setminus \mathrm{supp}(q_{\sigma_{k+1}})$ (refer to Figure 3). This means that a change in noise level suddenly causes the energy function to be ill-defined on a portion of the manifold of policy-generated motions, leading to poor rewards for these transitions and the introduction of non-stationarity in the problem. This can be verified with the energy return plot, where the return often drops at noise level changes indicating that the changed noise level is suddenly low-rewarding (Figure 5). Degradation at higher noise levels highlights the sensitivity of NEAR to the noise scale and is a limitation of this framework. Improvements can perhaps be made by experimenting with a linear noise scale, even more noise levels, or a more dynamic form of annealing.

Interestingly, ablation studies show that the addition of a task reward reduced the overall unpredictability of NEAR at the later stages of training. A reason for this could be the reduction in non-stationarity by the addition of a fixed component to the reward function. Finally, state-only reward learning techniques like NEAR and AMP are also quite sensitive to the motion dataset. We find that the policy is prone to converging to locally optimal behaviour if the dataset contains bi-directional state transitions (such that $(s, s')$ and $(s', s)$ are equally likely to occur).

Despite these limitations, the fundamental idea of using energy functions as reward functions still seems quite promising. Improvements can be made in several directions in the future. For example, it might be interesting to explore the effects of different noise distributions (instead of Gaussian noise) in the NCSN component of NEAR. Improvements to the annealing strategy could perhaps also be made by introducing softer noise-level changing criteria. One interesting idea could be to only use the updated reward function (post-annealing) for the portion of the transitions that are within the support of the new energy function and maintain the old reward function for other transitions. This might greatly reduce the negative effects of annealing shown in Figure 5. Other reinforcement learning improvements could also be made through ideas like targeted exploration, domain randomisation, or by changing the action sampling distribution from being Gaussian to a different probability distribution (Eberhard et al., 2023).

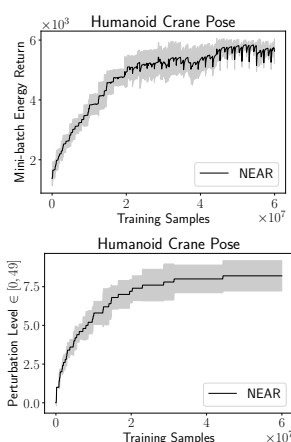

Figure 5: Annealing at higher noise levels causes a drop in the energy reward's return.

To conclude, this paper proposes an energy-based framework for imitation learning in partially observable conditions. Our framework builds on Noise-conditioned Score Networks (Song & Ermon, 2019) to explicitly learn a series of smooth energy functions from a dataset of expert demonstration motions. We propose to use these energy functions as reward functions to learn imitation policies via reinforcement learning. Further, we propose an annealing framework to gradually change the learnt reward functions, thereby providing a more focused and well-defined reward signal to the agent. Our proposed imitation learning algorithm called Noise-conditioned Energy-based Annealed Rewards (NEAR) outperforms state-of-the-art methods like Adversarial Motion Priors (AMP) in several quantitative metrics as well as qualitative evaluation across a series of complex contact-rich human imitation tasks.

### ACKNOWLEDGMENTS

The authors acknowledge the use of computational resources of the DelftBlue supercomputer provided by Delft High Performance Computing Centre (https://www.tudelft.nl/dhpc). The data used in this project was obtained from (i) mocap.cs.cmu.edu (created with funding from NSF EIA-0196217) and (ii) mocap.cs.sfu.ca (created with funding from NUS AcRF R-252-000-429-133 and SFU President's Research Start-up Grant).

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

# A PROOFS & EXTENDED EXPLANATIONS

## A.1 PROOF OF ENERGY FUNCTION SMOOTHNESS

In this section, we prove that an energy function learnt via denoising score matching (Vincent, 2011) is smooth and well-defined in the manifold of perturbed samples.

**Lemma A.1.** *Let $p_D$ be a distribution with support contained in a closed manifold $\mathcal{M} \subseteq \mathbb{R}^d$. We assume that $p_D$ is continuous in this manifold. Let $q_\sigma$ be a distribution supported in a closed manifold $\mathcal{P} \subseteq \mathbb{R}^d$ obtained by the addition of Gaussian noise $\mathcal{N}(x, \sigma) \triangleq (\exp -E_\sigma(x))/Z$ to each sample $x$ in $p_D$ (s.t. $q_\sigma(x) = \int \mathcal{N}(x'|x, \sigma I) p_D(x) dx$). Then $q_\sigma$ is continuous in $\mathcal{P}$ and $\nabla_x \log q_\sigma(x) = -\nabla_x E_\sigma(x)$ is smooth in $\mathcal{P}$.*

*Proof.* The convolution of a function with a Gaussian kernel results in a smooth function. By the same reasoning, $\nabla_x \log q_\sigma(x)$ is differentiable in $\mathcal{P}$ because $q_\sigma$ is continuous in $\mathcal{P}$. □

Lemma A.1 shows that a perturbed distribution and its score function are both smooth in the manifold of perturbed samples. Given this perturbed distribution, denoising score matching aims to learn a score function $s(x, \sigma) = \nabla_x e_\theta(x, \sigma)$ where $e_\theta(x, \sigma) \approx E_\sigma(x)$. From the universal approximation theorem (Cybenko, 1989; Hornik et al., 1989), it follows that a sufficiently large neural network can approximate any continuous function (score function in our case) on a compact domain to arbitrary precision, meaning that $\nabla_x e_\theta(x, \sigma)$ is a smooth function.

**Theorem A.2.** *Given a distribution $q_\sigma$ that is supported in a closed manifold $\mathcal{P}$ and is also continuous in this manifold, a parameterised energy function learnt via denoising score matching on samples drawn from $q_\sigma$ is smooth in $\mathcal{P}$.*

*Proof.* Lemma A.1 implies that the gradient of the score function is smooth in $\mathcal{P}$ and can be approximated smoothly by a neural network $\nabla_x e_\theta(x, \sigma)$. Since a function with a continuous gradient in a domain is itself continuous in that domain, it follows that $e_\theta(x, \sigma)$ is also continuous in $\mathcal{P}$. □

Continuity of $e_\theta(x, \sigma)$ in the whole space requires $\mathcal{P}$ to be equivalent to $\mathbb{R}^d$. However, the annealing strategy introduced in this paper and a sufficiently large $\sigma$ ensure that the manifold of policy-generated samples always lies in $\mathcal{P}$.

## A.2 ANNEALING DISCUSSION

### A.2.1 WHY ANNEAL IF THE LEARNT ENERGY FUNCTION IS SMOOTH?

If the learnt energy function $e_\theta(., \sigma_k)$ is smooth, then simply maximising it should still provide an unambiguous improvement signal to the agent. Why then must we anneal the energy functions?

Annealing not only ensures that the reward signal is well-defined, but it also progressively provides more "focused" rewards to the agent. Given a score function $s(x', \sigma) = \nabla_{x'} e_\theta(x', \sigma) = (x' - x)/\sigma^2$, for any fixed sample $x'$, $\nabla_{x'} e_\theta(x', \sigma_{k+1}) > \nabla_{x'} e_\theta(x', \sigma_k)$. This means that at any given point in training, the increase in received rewards for making positive progress is much higher for the $(k+1)^{\text{th}}$ energy function. This can greatly incentivise the agent to move closer to $p_D$.

### A.2.2 WHY DO SOME TASKS SUBSTANTIALLY BENEFIT FROM ANNEALING?

Ablation experiments from Section 5.3 show that annealing has a significant positive impact on more challenging tasks like crane pose. We hypothesise that the expert data distribution $p_D$ is more densely distributed for some tasks while sparsely for others (Figure 6). Assuming that a newly initialised policy always starts in the same manifold, the agent would start at a more informative reward function for a sparsely distributed $p_D$ than for a densely distributed $p_D$. This means that a noise level change for a densely distributed $p_D$ provides more informative rewards.

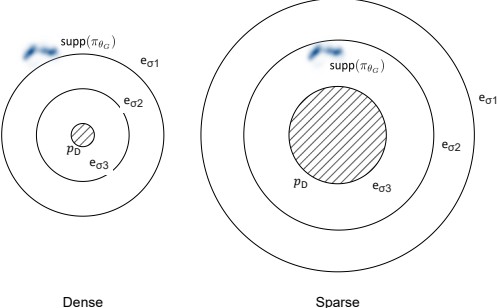

Figure 6: An illustration of a dense and sparse $p_D$. In the case of the sparse distribution, annealing would have a lower impact as the newly initialised policy would start out receiving much higher rewards.

## B EXPERIMENT DETAILS

### B.1 TASKS

The task reward and goal features for each imitation task are described below.

**Target Reaching**

In this task, the agent's objective is to navigate towards a randomly placed target. The agent's state is augmented to include a goal $g_t = x_t^*$ where t is the current timestep, and $x_t^*$ is the target's position in the agent's local coordinate frame. During training, the target is randomly initialised in a $240°$ arc around the agent within a radius in the range [2.0, 6.0] meters. The agent is rewarded for minimizing its positional error norm and heading error to the target. Here, $x_t$ is the agent's position, $v_t$ is the agent's velocity vector, $d^*_t$ is a unit vector pointing from the agent's root to the target, and $v^*$ is a scalar desired velocity set to 2.0 for walking and 4.0 for runninng.

$$r^{task} = 0.6 \left( \exp\left(-0.5 \left\| x_t^* - x_t \right\|^2 \right) \right) + 0.3 \left( 1 - \frac{2}{1 + \exp\left(5 * \frac{(v_t * d^*_t)}{\|v_t\|}\right)} \right) + 0.1 \left( 1 - (\|v_t\| - v^*)^2 \right)$$

(4)

**Target Reaching & Punching**

In this task, the agent's objective is to both reach a target and then strike it with a left-handed punch. Here, the goal is a vector of the target's position in the agent's local frame and a boolean variable indicating whether the target has been punched, $g_t = < x_t^*, \texttt{punch state} >$. We use the same target initialisation strategy as before with an arc of $45°$ and an arc radius in [1.0, 5.0] meters. The agent is rewarded using the target location reward when it is farther than a threshold distance from the target and with a target striking reward when it is within this threshold. The striking reward aims to minimise the pose error and heading error between the agent's end effector and the target while simultaneously aiming to achieve a certain end effector velocity and height. The complete reward function is shown below where $x_t^{eff}$ is the end effector position, $v_t^{eff}$ is the end effector velocity vector, $h_t^{eff}$ is the end effector height, and $v^{*eff} = 4.0$ and $h^{*eff} = 1.4$ are scalar desired punch speed and height.

$$r^{task} = \begin{cases} 1.0 & \text{target has been hit} \\ r^{near} & \|x_t - x_t^*\| < 1.2 \\ r^{far} & \text{otherwise} \end{cases}$$

(5)

$$r^{far} = \text{Equation (4)}$$

$$r^{near} = 0.3 + 0.3\left(0.1\left(\exp\left(-2.0\left\|x_t^* - x_t^{eff}\right\|^2\right)\right)\right) + 0.4\left(1 - \frac{2}{1 + \exp\left(5 * \frac{(v_t^{eff} * d^*{}_t)}{\left\|v_t^{eff}\right\|}\right)}\right)$$

$$+ 0.3\left(1 - \left(\left\|v_t^{eff}\right\| - v^{*eff}\right)^2\right) + 0.2\left(1 - (h_t^{eff} - h^{*eff})^2\right) \tag{6}$$

**Unconditioned Rewards**

These reward functions were used in our ablation experiments. In this case, we do not use any additional goal-conditioning.

*Walking & Running:* The agent is rewarded positively for every time step in the episode, encouraging longer episode lengths. Further, the agent is also rewarded for relative positive displacement, with the clipping at 0.5 meters. The final task reward is a weighted combination of both. $r_t^{task} = 0.5 \cdot 1 + 0.5 \cdot \frac{\texttt{clip}(x_t - x_{t-1}, 0.0, 0.5)}{0.5}$.

*Crane Pose & Punching:* The agent is rewarded positively for every time step in the episode, encouraging longer episode lengths. $r_t^{task} = 1$.

## B.2 TRAINING & EVALUATION DETAILS

### B.2.1 ARCHITECTURES

For both NEAR and AMP, the policy is a simple feed-forward neural network that maps the agent's state $s$ to a Gaussian distribution over actions, $\pi_{\theta_G} = \mathcal{N}(\mu(s), \Sigma)$ with the mean $\mu(s)$ being returned by the neural network and a fixed diagonal covariance matrix $\Sigma$. In our experiments, the neural network is a fully-connected network with (1024, 512) neurons and ReLU activations. $\Sigma$ is set to have values of $e^{-2.9}$ and stays fixed throughout training. The critic (value function) is also modelled by a similar network. The value function is updated with TD($\lambda$) (Sutton, 1988) and advantages are computed using generalised advantage estimation (Schulman et al., 2015). When using the environment-supplied task reward, we set $w^{task} = w^{energy} = 0.5$.

The NCSN neural network is a fully-connected network with an auto-encoder style architecture. Here, the encoder has (512, 1024) neurons and maps the input to a 2048-dimensional latent space. The decoder has (1024, 512, 128) neurons with the output being the unconditional energy of a sample. We use ELU activations between all layers of the auto-encoder and use Xavier uniform weight initialisation (Glorot & Bengio, 2010) to improve consistency across different independent training runs. Further, we standardise samples before passing them to the network. The NCSN noise scale was defined as a geometric sequence with $\sigma_1 = 20$, $\sigma_L = 0.01$, and $L = 50$ following the advice from Song & Ermon (2020). Following Song & Ermon (2020) we also track the exponentially moving average (EMA) of the weights of the energy network during training and use the EMA weight during inference, as it has been shown to further reduce the instability in the sample quality. All models in this paper were trained on the Nvidia-A100 GPU (Choquette et al., 2021).

### B.2.2 REINFORCEMENT LEARNING

We borrow the experimental setup from Peng et al. (2021) where the agent's state is a 105-dimensional vector consisting of the relative position of each link with respect to the root body and the rotation of each link (represented as a 6-dimensional normal-tangent vector of the link's linear and angular velocities). All features are in the agent's local coordinate system. Similarly to Peng et al. (2021), we do not add additional features to encode information like the feature's phase in the motion, or the target pose. Further, the character is not trained to replicate the phase-wise features of the motion and the learnt rewards are generally only a representation of the closeness of the agent's motion to the expert's data distribution. The agent's actions specify a positional target that is then tracked via PD controllers at each joint.

We use asynchronous parallel training in the IsaacGym simulator (Makoviychuk et al., 2021) for all experiments and analyses in this paper and spawn 4096 independent, parallel environments during both training and evaluation. Given an initial policy, training is carried out by first rolling out the policy in all environments for a rollout horizon (16 in our experiments). During rollouts, an environment is reset if it happens to reach the `done` state. A replay buffer is then populated with all agents' transitions obtained from the rollouts. Then, with rollouts paused, the policy and value function are updated with the data from the replay buffer. After the update, the rollouts are restarted with the updated policy. We use multiple mini epochs to update the policy and value function at every update step. In each mini epoch, several mini-batches of data samples are drawn from the replay buffer. The number of mini-batches depends on the relative size of each mini-batch and the replay buffer.

During policy evaluation, the same rollout procedure is used, however this time, the rollout horizon is set to 300 and the networks are of course not updated. Performance metrics are recorded every main training epoch as a mean across the $k = 20$ most rewarding environments. Finally, we use reference state initialisation to initialise all environments at a random state in the expert's motion dataset and use early termination to reset when the agent falls over. For certain tasks like spin-kick, we find that it is especially challenging to learn a policy starting from certain initial states (such as the jumping-off point). Additional exploration is required to learn the optimal actions starting from these states. In these cases, reference states are drawn from a beta distribution instead of a uniform distribution (with $\beta = 3.0$ and $\alpha = 1.0$). For temporally composed tasks like target reaching and punching, we use both reference motions during initialisation. In this case, both reference motions are used with a probability of 0.5 and the target is initialised in the range [1.0, 1.2] when the agent is initialised with the punching reference.

### B.2.3 Evaluation Metrics

*Average Dynamic Time Warping Pose Error:* This is the mean dynamic time warping (DTW) error (Sakoe & Chiba, 1978) between trajectories of the agent's and the expert's poses averaged across all expert motions in the dataset. Given a set of $j$ expert motion trajectories $\hat{\tau}_j$ of arbitrary length $L_j$ where each trajectory is a series containing the Cartesian positions of the reference character's joints $\hat{x}_i$, $D = \{\hat{\tau}_j\}_{j=1}^{N_{traj}}$ where $\hat{\tau}_j = \{\hat{x}_i\}_{i=1}^{L_j}$, first, we roll out the trained policy deterministically [6] across several thousand random starting-pose initialisations [7]. Then, the $k = 20$ most rewarding trajectories are selected to form a set of policy trajectories $D_\pi = \{\tau_m\}_{m=1}^k$ where $\tau_m = \{x_i\}_{i=1}^{L_m}$ and each trajectory has an arbitrary length $L_m$. The average dynamic time warping pose error is then computed as the average DTW score of all $\tau_m$ across all expert trajectories $\hat{\tau}_j$ with $\|\hat{x}_i - x_i\|_2$ as the cost function. To ensure that the pose error is only in terms of the character's local pose and not its global position in the world, we transform each Cartesian position to be relative to the character's root body position at that timestep ($\hat{x}_i \leftarrow \hat{x}_i - \hat{x}_i^{root}$ and $x_i \leftarrow x_i - x_i^{root}$).

*Spectral Arc Length:* Spectral Arc Length (SAL) (Beck et al., 2018; Balasubramanian et al., 2011; 2015) is a measure of the smoothness of a motion. The smoothness of the character's trajectory is an interesting metric to determine the policy's ability to perform periodic motions in a controlled manner. The underlying idea behind SAL is that smoother motions typically change slowly over time and are comprised of fewer and low-valued frequency domain components. In contrast, jerkier motions have a more complex frequency domain signature that consists of a lot of high-frequency components. The length of the frequency domain signature of a motion is hence an appropriate indication of a motion's smoothness (with low values indicating smoother motions). SAL is computed by adding up the lengths of discrete segments (arcs) of the normalised frequency-domain map of a motion. In our experiments, we use SPARC Beck et al. (2018), a more robust version of the spectral arc length that is invariant to the temporal scaling of the motion. We track the average SAL of the $k = 20$ most rewarding trajectories generated by the policy at different training intervals and use the root body Cartesian coordinates to compute the SAL. Note that in this case, we do not transform the positions to the agent's local coordinate system.

---

[6]Deterministic here means that we use the predicted mean as the action instead of sampling from $\mathcal{N}(\mu(x), \Sigma)$.

[7]This is done to ensure that the computed performance is not biased to any single initial state.

### B.2.4 HYPERPARAMETERS & TRAINING ITERATIONS

Table 4: Hyperparameters used in our experiments. Architectural details are mentioned in Appendix B.2

| Hyperparam | Value |
|---|---|
| ***Reinforcement Learning*** | |
| Discount Factor $\gamma$ | 0.99 |
| GAE $\lambda$ | 0.95 |
| TD $\lambda$ | 0.95 |
| Learning Rate | 5 e-5 |
| PPO clip threshold | 0.2 |
| Training horizon | 16 |
| Mini-batch size | 2048 |
| | |
| ***NEAR (Energy NCSN)*** | |
| Batch size | 128 |
| $\sigma_1$ | 20 |
| $\sigma_L$ | 0.01 |
| Num noise levels L | 50 |
| Exponentially moving avg. rate | 0.999 |
| Learning rate | 1 e-5 |
| Adam - $\beta$ | 0.9 |
| | |
| ***AMP (Discriminator)*** | |
| Batch size | 512 |
| Gradient penalty | 5 |
| Demo observations buffer size | 2 e5 |
| Discriminator loss coefficient | 5 |
| Discriminator output regularisation | 0.05 |

Table 5: Details of NCSN training iterations (NEAR only) and reinforcement learning environment interactions (same for NEAR & AMP). [†]: clips include turning motions. [‡]: clips include turning and punching motions.

| Task | Num. Motion Clips | NCSN Iters. | RL Env. Interactions |
|---|---|---|---|
| Walking | 74 | 1.5 e5 | 60 e6 |
| Running | 26 | 1.5 e5 | 60 e6 |
| Crane Pose | 3 | 1.0 e5 | 60 e6 |
| Left Punch | 19 | 1.2 e5 | 80 e6 |
| Mummy Walk | 1 | 0.8 e5 | 80 e6 |
| Spin Kick | 1 | 1.2 e5 | 100 e6 |
| | | | |
| Target Reaching (walking) | 22[†] | 1.5 e5 | 100 e6 |
| Target Reaching (running) | 8[†] | 1.2 e5 | 100 e6 |
| Target Reaching & Punching | 33[‡] | 1.2 e5 | 100 e6 |

### B.2.5 REPEATABILITY & DETERMINISM

Each algorithm was trained 5 times independently on every task with separate random number generator seeds for each run. However, using a fixed seed value will only potentially allow for deterministic behaviour in the IsaacGym simulator. Due to GPU work scheduling, it is possible that runtime changes to simulation parameters can alter the order in which operations take place, as environment updates can happen while the GPU is doing other work. Because of the nature of floating point numeric storage, any alteration of execution ordering can cause small changes in the least significant bits of output data, leading to divergent execution over the simulation of thousands of environments and simulation frames. This means that experiments from the IsaacGym simulator (including the original work on AMP) are not perfectly reproducible on a different system. However, parallel simulation is a major factor in achieving the results in this paper and minor non-determinism between independent runs is hence just an unfortunate limitation. More information on this can be found in the IsaacGymEnvs benchmarks package. Note that this is only a characteristic of the reinforcement learning side of our algorithm. The pretrained energy functions are also seeded and these training runs are perfectly reproducible.

## B.3    MAZE DOMAIN DETAILS

This section provides additional details of the experiments and procedures used to generate Figure 1. In this experiment, the agent is initialised randomly in a small window at the top portion of the L-shaped maze. The agent aims to reach the goal position at the bottom right (the episode ends when the agent's position is within some threshold of the goal). Expert demonstrations were collected, so the expert's trajectory did not reach the target directly but first passed through an L-shaped maze. The agent is expected to learn to imitate this by passing through the maze. We train AMP and NEAR in this domain and visualise the learnt reward functions. In the case of NEAR, we visualise the energy function ($e_\theta(\cdot, \sigma)$ with $\sigma = 20.0$) and in the case of AMP we visualise the discriminator. The energy function is trained by training our modified NCSN on the expert state transitions in the maze domain. The discriminator is trained while training the AMP policy. Figure 1 shows two comparisons. We compare $\texttt{rew}(s'|s)$ at a fixed state $s$ in the maze at different training iterations. The energy-based reward function is stationary throughout training and $\texttt{rew}(s'|s)$ only changes with $s'$ for a fixed $s$. In contrast, the adversarial reward depends on the agent's policy and keeps changing to minimise the prediction for the samples in the policy's distribution $p_G$.

There is no environment-provided reward function in this domain and the visualisations are obtained only by using the learnt rewards. Apart from visualisation and domain-related differences, the training regime for both NEAR and AMP in this task is identical to the training regime used in all other experiments in this paper. Please refer to Appendix B.2 for training details.

## C    EXTENDED RESULTS

Table 6: A comparison of the mean performance of NEAR and AMP at the end of training (*Avg. pose error:* lower is better. *Others:* closer to expert is better). Stdev. across independent runs is shown as an error value ($\pm$).

| Task | Algorithm | Avg. Pose Error (m) | Spectral Arc Length | Root Body Velocity ($\frac{m}{s}$) | Root Body Jerk ($\frac{m}{s^3}$) |
|---|---|---|---|---|---|
| Walking | NEAR | **0.51** ± **0.15** | **-7.52** ± **1.32** | **1.25** ± **0.18** | **360.89** ± **184.36** |
| | AMP | **0.51** ± **0.07** | -8.78 ± 1.04 | 1.87 ± 0.1 | 736.32 ± 78.25 |
| | Expert | - | -5.4 | 1.31 | 130.11 |
| Running | NEAR | **0.62** ± **0.17** | **-7.24** ± **1.59** | **3.52** ± **0.37** | **1298.42** ± **215.42** |
| | AMP | 0.65 ± 0.01 | -9.71 ± 1.54 | 3.79 ± 0.14 | 1560.14 ± 87.18 |
| | Expert | - | -3.79 | 3.55 | 513.68 |
| Crane Pose | NEAR | 0.94 ± 0.15 | -6.6 ± 1.97 | 0.12 ± 0.22 | **46.77** ± **77.72** |
| | AMP | **0.82** ± **0.09** | **-8.1** ± **1.18** | **0.03** ± **0.01** | 19.29 ± 5.17 |
| | Expert | - | -12.28 | 0.03 | 49.05 |
| Left Punch | NEAR | 0.37 ± 0.05 | **-6.87** ± **1.47** | 0.01 ± 0 | 11.34 ± 2.86 |
| | AMP | **0.32** ± **0.01** | -9.93 ± 3.28 | **0.06** ± **0.02** | **29.6** ± **8.16** |
| | Expert | - | -1.73 | 0.16 | 72.49 |
| Mummy Walk | NEAR | 0.66 ± 0.39 | **-4.72** ± **1.2** | 0.33 ± 0.4 | **189.73** ± **189.41** |
| | AMP | **0.41** ± **0.01** | -13.84 ± 1.12 | **0.98** ± **0.04** | 354.49 ± 33.02 |
| | Expert | - | -4.71 | 0.73 | 79.63 |
| Spin Kick | NEAR | 0.78 ± 0.05 | -5.59 ± 2.26 | **0.53** ± **0.19** | 286.63 ± 60.77 |
| | AMP | **0.58** ± **0.1** | **-3.16** ± **0.73** | 0.5 ± 0.14 | **278.25** ± **29.52** |
| | Expert | - | -3.39 | 1.05 | 273.61 |

Table 7: A comparison of wall-clock times of NEAR and AMP. Stdev. across independent runs is shown as an error value ($\pm$). We find that overall NEAR requires slightly more training time than AMP. Across all tasks, NCSN contributes to less than 30% of the total computational time of NEAR, with RL accounting for the majority of the computation load and wall-clock time. Interestingly, AMP always has a larger reinforcement learning computational load. This is expected as AMP learns both the policy and the reward function simultaneously.

| Task | Algorithm | Avg. NCSN Wall Time (min.) | Avg. RL Wall Time (min.) | Total Wall Time (min.) [8] |
|---|---|---|---|---|
| Walking | NEAR | $10.19 \pm 0.22$ | $\mathbf{15.07} \pm \mathbf{0.23}$ | $25.26 \pm 0.32$ |
| | AMP | - | $18.46 \pm 0.52$ | $\mathbf{18.46} \pm \mathbf{0.52}$ |
| Running | NEAR | $8.11 \pm 0.3$ | $\mathbf{15.15} \pm \mathbf{0.29}$ | $23.26 \pm 0.42$ |
| | AMP | - | $17.97 \pm 0.42$ | $\mathbf{17.97} \pm \mathbf{0.42}$ |
| Crane Pose | NEAR | $6.61 \pm 0.16$ | $\mathbf{13.92} \pm \mathbf{0.6}$ | $20.53 \pm 0.62$ |
| | AMP | - | $17.16 \pm 0.5$ | $\mathbf{17.16} \pm \mathbf{0.5}$ |
| Left Punch | NEAR | $7.83 \pm 0.17$ | $\mathbf{20.39} \pm \mathbf{0.42}$ | $28.22 \pm 0.45$ |
| | AMP | - | $22.42 \pm 0.61$ | $\mathbf{22.42} \pm \mathbf{0.61}$ |
| Mummy Walk | NEAR | $5.07 \pm 0.1$ | $\mathbf{18.11} \pm \mathbf{0.32}$ | $23.18 \pm 0.34$ |
| | AMP | - | $21.13 \pm 0.33$ | $\mathbf{21.13} \pm \mathbf{0.33}$ |
| Spin Kick | NEAR | $7.73 \pm 0.2$ | $\mathbf{19.94} \pm \mathbf{0.46}$ | $27.67 \pm 0.5$ |
| | AMP | - | $24.47 \pm 0.42$ | $\mathbf{24.47} \pm \mathbf{0.42}$ |
| Target Reaching (walking) | NEAR | $10.52 \pm 0.2$ | $\mathbf{28.93} \pm \mathbf{0.65}$ | $39.45 \pm 0.68$ |
| | AMP | - | $36.65 \pm 0.22$ | $\mathbf{36.65} \pm \mathbf{0.22}$ |
| Target Reaching (running) | NEAR | $8.08 \pm 0.14$ | $\mathbf{28.71} \pm \mathbf{0.86}$ | $36.79 \pm 0.87$ |
| | AMP | - | $34.95 \pm 1.39$ | $\mathbf{34.95} \pm \mathbf{1.39}$ |
| Target Reaching & Punching | NEAR | $7.93 \pm 0.09$ | $\mathbf{28.68} \pm \mathbf{0.83}$ | $\mathbf{36.61} \pm \mathbf{0.83}$ |
| | AMP | - | $37.54 \pm 0.28$ | $37.54 \pm 0.28$ |

## D    EXTENDED ABLATIONS

Table 8: A comparison of ablated configurations of NEAR (*Avg. pose error:* lower is better. *Others:* closer to expert is better). Stdev. across independent runs is shown as an error value ($\pm$).

| Task | Config | Avg. Pose Error (m) | Spectral Arc Length | Root Body Velocity ($\frac{m}{s}$) | Root Body Jerk ($\frac{m}{s^3}$) |
|---|---|---|---|---|---|
| Walking | anneal & $e_\theta$ | $0.51 \pm 0.15$ | $-7.52 \pm 1.32$ | $1.25 \pm 0.18$ | $360.89 \pm 184.36$ |
| | anneal & $\tilde{r}$ | $0.42 \pm 0.02$ | $-6.11 \pm 2.01$ | $0.88 \pm 0.58$ | $249.97 \pm 136.56$ |
| | $\sigma_5$ & $e_\theta$ | $0.49 \pm 0.22$ | $-7.1 \pm 1.94$ | $1.58 \pm 0.38$ | $653.17 \pm 554.85$ |
| | $\sigma_5$ & $\tilde{r}$ | $0.42 \pm 0.02$ | $-8.7 \pm 1.2$ | $1.25 \pm 0.51$ | $360.86 \pm 154.68$ |
| | Expert | - | $-5.4$ | $1.31$ | $130.11$ |
| Running | anneal & $e_\theta$ | $0.62 \pm 0.17$ | $-7.24 \pm 1.59$ | $3.52 \pm 0.37$ | $1298.42 \pm 215.42$ |
| | anneal & $\tilde{r}$ | $0.59 \pm 0.03$ | $-8.02 \pm 0.57$ | $4.86 \pm 0.16$ | $1875.44 \pm 72.8$ |
| | $\sigma_5$ & $e_\theta$ | $0.62 \pm 0.18$ | $-6.69 \pm 1.85$ | $3.48 \pm 0.52$ | $1334.1 \pm 244.91$ |
| | $\sigma_5$ & $\tilde{r}$ | $0.57 \pm 0.04$ | $-8.61 \pm 0.86$ | $4.76 \pm 0.1$ | $1826.98 \pm 57.54$ |
| | Expert | - | $-3.79$ | $3.55$ | $513.68$ |
| Crane Pose | anneal & $e_\theta$ | $0.94 \pm 0.15$ | $-6.6 \pm 1.97$ | $0.12 \pm 0.22$ | $46.77 \pm 77.72$ |
| | anneal & $\tilde{r}$ | $1.33 \pm 0.21$ | $-7.24 \pm 2.51$ | $0.14 \pm 0.23$ | $72.24 \pm 109.83$ |
| | $\sigma_5$ & $e_\theta$ | $1.38 \pm 0.8$ | $-6.03 \pm 1.79$ | $0.07 \pm 0.1$ | $29.5 \pm 40.22$ |
| | $\sigma_5$ & $\tilde{r}$ | $1.23 \pm 0.07$ | $-4.34 \pm 1.02$ | $0.02 \pm 0.01$ | $16.3 \pm 4.65$ |
| | Expert | - | $-12.28$ | $0.03$ | $49.05$ |

## E    ADVERSARIAL IL CHALLENGES & AMP DISCRIMINATOR EXPERIMENTS

In this section, we elaborate on the challenges of adversarial imitation learning and provide additional empirical results demonstrating instability and non-smoothness in the AMP discriminator. As briefly highlighted in Section 2, the root causes for the challenges of adversarial techniques are the

---

[8] For NEAR, the total wall time is calculated by assuming that the wall times of NCSN and RL runs are normally distributed. Total wall time = NCSN avg. + RL avg. $\pm \sqrt{\text{NCSN std.}^2 + \text{RL std.}^2}$

simultaneous min-max optimisation in their training procedure and the formulation of the discriminator as a classifier. Throughout training, the policy is updated to bring $p_G$ closer to $p_D$, meaning that the support of $p_G$ – the manifold of the samples generated by the policy – keeps changing. Simultaneously, the discriminator's decision boundary is also constantly changing to distinguish between the samples in $\text{supp}(p_D)$ and $\text{supp}(p_G)$. This iteratively changing adversarial procedure leads to high-variance discriminator predictions and causes performance instability.

At any point in training, the discriminator is trained to discriminate $p_D$ from $p_G$ – and *not* $p_D$ from all that is not $p_D$ – meaning that it is quite accurate on samples in $\text{supp}(p_D)$ and $\text{supp}(p_G)$ but is often arbitrarily defined in other regions of the sample space (Arjovsky & Bottou, 2017). The changing nature of $\text{supp}(p_G)$ means that after a policy update, some of the samples passed on to the discriminator could potentially have come from a region outside these two supports. Since the discriminator is arbitrarily defined here, it is likely to return misleading predictions, leading to misleading policy updates. This variance is potentially further heightened by the stochastic nature of reinforcement learning techniques like Proximal Policy Optimisation (PPO) (Schulman et al., 2017), meaning that the agent's exploration is typically met with poor rewards [9]. We empirically verify this high-variance hypothesis through experiments on AMP (Appendix E.1.1).

Further, reward smoothness is indeed an important criterion for faster convergence and sensible policy improvements in RL methods like PPO. An ideal data-driven reward function is both smooth in the sample space as well as consistent throughout training (stationarity). Unfortunately, because the discriminator $D_{\theta_D}$ is non-smooth in the whole space $X$ and is arbitrarily defined in $(\text{supp}(p_D) \cup \text{supp}(p_G))^\mathsf{c}$ – parts of the sample space that are unexplored and not in the demonstration dataset $\mathcal{M}$ –, the rewards in this region are also non-smooth. Moreover, the iteratively changing nature of the discriminator's decision boundary means that the reward function is also non-stationary. These issues compound such that the agent often receives constant or arbitrarily changing rewards and hence the policy receives uninformative updates. Appendix E.1.2 discusses experiments that highlight this non-smoothness and non-stationarity in AMP.

Finally, adversarial learning techniques are also prone to poor performance due to perfect discrimination. Arjovsky & Bottou (2017) introduce the perfect discriminator theorems that state that if $p_D$ and $p_G$ have disjoint supports or have supports that lie in low-dimensional manifolds (lower than the dimension of the sample space $X$), then there exists an optimal discriminator $D^* : X \rightarrow [0, 1]$ that has accuracy 1 and $\nabla_x D^*(x) = 0 \forall x \in \text{supp}(p_D) \cup \text{supp}(p_G)$ (Theorems 2.1 and 2.2 in (Arjovsky & Bottou, 2017)). They also prove that under these conditions $p_G$ is non-continuous in $X$ and it is increasingly unlikely that $\text{supp}(p_D)$ and $\text{supp}(p_G)$ perfectly align (Lemmas 2 and 3 in (Arjovsky & Bottou, 2017)). In the case of AIL, the generator is indeed a neural network mapping low-dimensional samples (features $s \in S$) to the discriminator's high dimensional input space. Further, $p_G$ and $p_D$ are potentially disjoint and it is at least unlikely that their supports perfectly align. This means that at the initial stages of training, the discriminator very quickly learns to perfectly distinguish between the samples in the expert dataset $\mathcal{M}$ and those in $p_G$, assigning a prediction of 0 to any sample in the agent's trajectory. When used as a reward function, $\log D_{\theta_D}(W(\pi_{\theta_G}(s))) = \log 0$, instantly leads to arbitrary policy updates. Even when using a modified reward formulation, say $D()$ instead of $\log D()$, the agent would receive a nearly constant reward, say $c$. Under such a constant reward function, the gradient of the performance measure quickly goes down to zero – since the expectation of the gradient of the log probability of a parameterised distribution (Fisher score) is zero at the parameter (Wasserman, 2013). We verify these claims by replicating the experiments from Arjovsky & Bottou (2017) on adversarial motion priors (AMP) (Appendix E.1.3).

To conclude this section we highlight that NEAR does not rely on such simultaneous optimisation and instead has a fixed, stationary reward function. Hence, NEAR is not prone to such instability and non-smoothness.

---

[9]We agree that the KL diverge constraint in PPO might somewhat reduce the negative impacts of this, however, penalising policy change with worse rewards is still non-ideal.

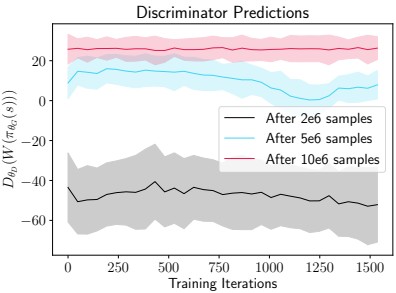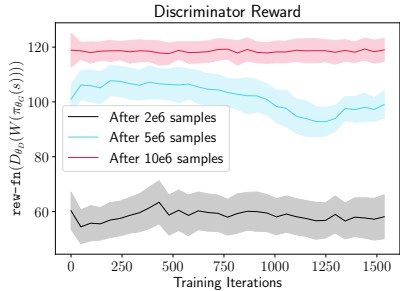

Figure 7: Discriminator variance experiments. The policy was first trained for $2e6$, $5e6$, and $10e6$ data samples. Then, with the policy updates paused, discriminator training was continued. We plot the rewards received across several independent policy rollouts and observe a high variance. Given that the policy is unchanging, a high variance in the reward indicates poor reinforcement learning.

### E.1 AMP DISCRIMINATOR EXPERIMENTS

#### E.1.1 HIGH DISCRIMINATOR VARIANCE

We conduct additional experiments to very the high variance in the adversarial motion priors (Peng et al., 2021) discriminator predictions. We train AMP on a humanoid walking task using the loss function from Equation (1) for the discriminator and Proximal Policy Optimisation (PPO) (Schulman et al., 2017) to train the policy. The discriminator is slightly modified by removing the sigmoid activation at the output layer and instead computing the loss on `sigmoid`$(D())$ [10] (same setup as the main experiments in this paper). Training is continued normally until some cut-off point. The cut-off point is varied across runs to obtain varying levels of intersection between $\text{supp}(p_D)$ and $\text{supp}(p_G)$. Then, with the policy updates paused, we continue training the discriminator to maintain the learnt decision boundary and visualise the variance in the trained discriminator's predictions on the motions generated by an unchanging policy. We hypothesise that as training continues and the supports of the two distributions get closer, the discriminator is less likely to see samples from a region outside $\text{supp}(p_D) \cup \text{supp}(p_G)$, meaning that its variance reduces as training progresses. We observe that the discriminator's predictions indeed have quite a high variance and the range of the predictions varies vastly across training levels (Figure 7). Further, the variance indeed reduces over the training level, indicating a gradually increasing intersection between $\text{supp}(p_D)$ and $\text{supp}(p_G)$. The adversarial optimisation is likely to get stabilised as the policy gets closer to optimality, however, training for the most part is still rather unstable because of the high variance in the discriminator's predictions.

#### E.1.2 DISCRIMINATOR NON-SMOOTHNESS

We also conduct experiments to understand the smoothness of the learnt reward function and its changes over training iterations (Figure 8). To do this, we again train AMP on a humanoid walking task. This time we do not modify the algorithm and simply evaluate the discriminator at gradually increasing distances from the true data manifold at various points in training. We find that the discriminator's predictions on average, decline as we move farther away from the true data manifold. However, again, the predictions are quite noisy and have a fairly large standard deviation.

#### E.1.3 PERFECT DISCRIMINATION

Finally, we replicate the experiments from Arjovsky & Bottou (2017) on adversarial IL (Figure 9). We use the same experimental setup as Appendix E.1.1 but now compute the accuracy with the output of the final Sigmoid layer. Here, instead of continuing the discriminator's training, we retrain the discriminator to distinguish between samples in the expert dataset $p_D$ and samples in $p_G$ (same

---

[10]This is done to allow the network to predict any arbitrary value and to allow more flexibility in the reward function transformation. The same is done in the original AMP procedure Peng et al. (2021) and we make no additional modifications to their code.

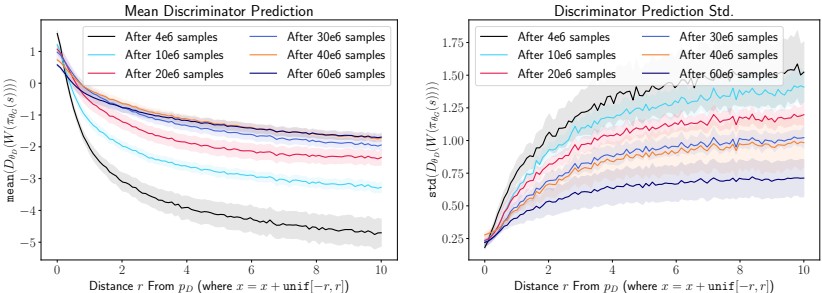

Figure 8: Discriminator non-smoothness experiments. Plots show the mean and std. discriminator prediction over a large batch of perturbed expert data samples at varying levels of perturbation (where a distance of 0 on the x-axis corresponds to unperturbed expert data in $p_D$). Notice the high value of the std. compared to the mean at any given distance from $p_D$.

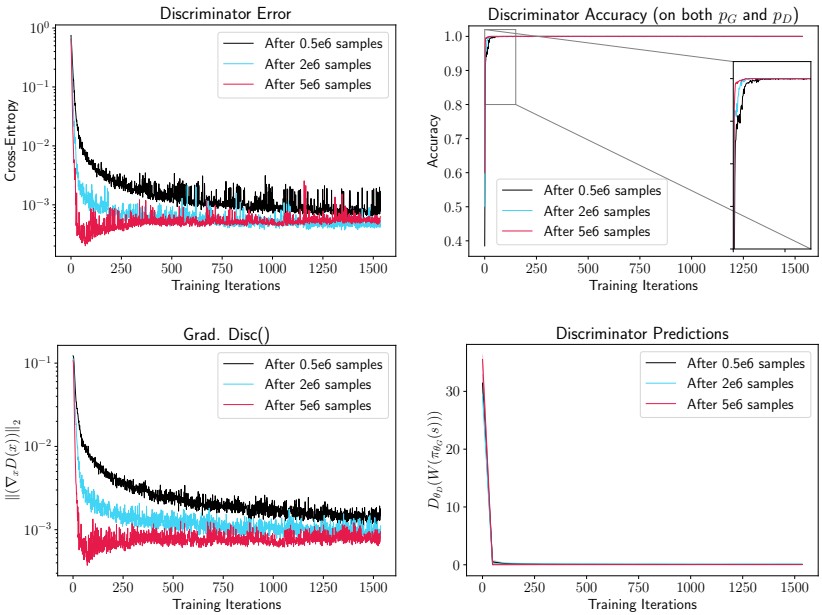

Figure 9: Perfect discriminator experiments on the walking task (multi-clip dataset with 74 motions clips). The policy was first trained for $0.5e6$, $2e6$, and $5e6$ data samples. Then, with the policy updates paused, the discriminator was retrained. We find that the discriminator very quickly learns to perfectly distinguish between $p_D$ and $p_G$ (notice the logarithmic scale).

procedure as Arjovsky & Bottou (2017)). Our experiments reproduce the results from Arjovsky & Bottou (2017) on adversarial IL (Figure 9). The discriminator loss from Equation (1) rapidly declines indicating a near-perfect discriminator prediction and highlighting the fact that even after sufficient training, $p_G$ and $p_D$ are non-continuous. The accuracy of the discriminator reaches a value of 1.0 in at most 75 iterations and $\nabla_x D(x)$ rapidly declines to be 0, further corroborating the theoretical results. Finally, we also find that the discriminator's predictions on the motions generated by the policy rapidly drop down to zero, meaning that the policy receives unhelpful updates.

Despite these core issues of AIL, it is still unclear why these techniques (and traditional GANs) still function comparably to score-based alternatives. One reason could be that the changing discriminator inputs mitigate the challenges of perfect discrimination. However, this is still an open question that needs future work and deeper analysis.

