# OpenReview forum: "Noise-conditioned Energy-based Annealed Rewards (NEAR): A Generative Framework for Imitation Learning from Observation"
_ICLR.cc/2025/Conference — ICLR 2025 Poster_

### Official Review · Reviewer_hjSn · 2024-10-19

**Soundness:** 3
**Presentation:** 3
**Contribution:** 3
**Rating:** 6
**Confidence:** 4

**Summary:**

This paper introduces Noise-conditioned Energy-based Annealed Rewards, a new framework for inverse reinforcement learning that leverages diffusion models. Instead of using unstable and non-smooth adversarial learning to approximate reward functions, NEAR learns an energy function via score matching. NEAR provides smooth and accurate reward signals for training policies through reinforcement learning.

**Strengths:**

NEAR replaces adversarial learning with energy-based modeling using diffusion models and score matching. This results in a stable and smooth reward representation, addressing issues of instability and non-smooth reward landscapes inherent in adversarial methods.

**Weaknesses:**

**Lack of Comparison with Related Work:** While the paper acknowledges limitations like stability with large noise levels and data requirements, a significant concern is the absence of direct comparison with other works that apply diffusion models to imitation learning from observation. Specifically, works like [1] and [2] also leverage diffusion models in inverse reinforcement learning. Since the authors claim that their work is the first to apply diffusion models for reward learning, providing further clarification (for instance, [2] is a general diffusion-based IRL algorithm, it could be helpful if the author could highlight the difference in the paper) and direct comparisons in the experiment section with these methods would greatly enhance the paper's quality and situate it within the existing literature.

[1] B. Wang, G. Wu, T. Pang, Y. Zhang, and Y. Yin, “DiffAIL: Diffusion Adversarial Imitation Learning,” Dec. 12, 2023, arXiv: arXiv:2312.06348. Accessed: Oct. 19, 2024. [Online]. Available: http://arxiv.org/abs/2312.06348

[2] R. Wu, Y. Chen, G. Swamy, K. Brantley, and W. Sun, “Diffusing States and Matching Scores: A New Framework for Imitation Learning,” Oct. 17, 2024, arXiv: arXiv:2410.13855. Accessed: Oct. 19, 2024. [Online]. Available: http://arxiv.org/abs/2410.138

**Questions:**

1. **Comparison with DiffAIL [1]:** [1] also applies inverse reinforcement learning using diffusion models. Could the authors explain NEAR's major advantages compared with other works that integrate diffusion models into IRL? For instance, as the authors mention that GAIL can work well with single-clip data, how does NEAR compare in such settings, especially regarding data efficiency and performance? Further justification or emperial evaluation would be appreciated.

2. **Comparison with SMILING [2]:** [2] introduces a non-adversarial framework using diffusion models for imitation from observation and provides theoretical analysis. Could the authors elaborate on the differences and advantages of NEAR compared with [2]? Including further empirical results comparing NEAR with [2] would strengthen the paper and clarify NEAR's contributions relative to existing methods.


[1] B. Wang, G. Wu, T. Pang, Y. Zhang, and Y. Yin, “DiffAIL: Diffusion Adversarial Imitation Learning,” Dec. 12, 2023, arXiv: arXiv:2312.06348. Accessed: Oct. 19, 2024. [Online]. Available: http://arxiv.org/abs/2312.06348

[2] R. Wu, Y. Chen, G. Swamy, K. Brantley, and W. Sun, “Diffusing States and Matching Scores: A New Framework for Imitation Learning,” Oct. 17, 2024, arXiv: arXiv:2410.13855. Accessed: Oct. 19, 2024. [Online]. Available: http://arxiv.org/abs/2410.138

---

> ### Author Response · Authors · 2024-11-16
> **Review responses & paper revisions**
>
> Dear reviewer, thank you for your feedback. We have revised the paper. Please find clarifications below. We hope these sufficiently establish the contribution of the paper and its place within the IL from the observation field. Note: some lines in the text are coloured to indicate suggested modifications to other reviewers. These will be turned back to normal after the rebuttals.
>
> > Comparison with DiffAIL [1]: [1] also applies inverse reinforcement learning using diffusion models. Could the authors explain NEAR's major advantages compared with other works that integrate diffusion models into IRL?
> - Thank you for pointing out the DiffAIL [1] paper. This is indeed an interesting paper. There are two main differences between NEAR and DiffAIL. First,  DiffAIL is a combination of diffusion and adversarial techniques. It relies on the same objective as older adversarial methods like GAIL (please refer to METHOD: DIFFUSION ADVERSARIAL IMITATION LEARNING in [1]) and adds the diffusion loss within this term (eq. 17 in [1]). Hence, it might still be prone to the challenges of adversarial optimisation that we try to eliminate in our paper.
> - Further, the use of the diffusion mechanism in DiffAIL is different to the use of score-based models in NEAR. In NEAR, we use score-based models to learn an energy function **prior** to learning the policy. We then use this fixed energy function to train the policy. Instead, DiffAIL trains the diffusion models simultaneously in hopes that it provides a more accurate reward than the simple adversarial classifier. Hence, while both algorithms use score-based models, their purpose in our opinion is quite different.
> - Finally, DiffAIL has significantly higher computational costs because of the added load of learning the diffusion-based classifier (Table 4 in [1]). In comparison NEAR only has a fraction of the computation load (Table 7 in our paper).
> - Having said that, we agree that it is important to mention this paper in ours. We have now added DiffAIL to our related work (line 72). We also plan to add this as a baseline in our future work on NEAR.
>
> > For instance, as the authors mention that GAIL can work well with single-clip data, how does NEAR compare in such settings, especially regarding data efficiency and performance? Further justification or empirical evaluation would be appreciated.
> - Thank you for pointing out the single-clip performance of NEAR. We also trained NEAR in data-limited single-clip settings. These results are provided in Table 1 and Figure 4 in the paper (mummy-style walking and spin-kick are single-clip tasks). From our experiments, we found that NEAR performs poorly in single-clip settings while AMP is less affected by data unavailability. We believe that this is because of the added challenge of accurately learning an energy function via score-based modelling in limited data.
> - We have now also provided additional information on the training resources (Table 5) and computational efficiency (Table 7) of NEAR. We find that in most cases (including single-clip settings), NEAR has better RL computational efficiency than AMP. Although, overall, it still requires slightly more training time.
> - We hope that this explanation and additional results clarify NEAR’s performance under limited data.
>
> >Comparison with SMILING [2]: [2] introduces a non-adversarial framework using diffusion models for imitation from observation and provides theoretical analysis. Could the authors elaborate on the differences and advantages of NEAR compared with [2]? Including further empirical results comparing NEAR with [2] would strengthen the paper and clarify NEAR's contributions relative to existing methods.
> - While SMILING [2] is indeed very interesting research, it is a recently proposed algorithm (submitted on ArXiV on 17 Oct 2024 which is after the ICLR 2025 submission date) and is currently also under peer review at ICLR. Hence, unfortunately, we could not include this in our literature search or empirical analysis.
>
> > A significant concern is the absence of direct comparison with other works that apply diffusion models to imitation learning from observation.
> - To the best of our knowledge, NEAR is novel in its use of score-based models and energy-based rewards for imitation learning from observation (excluding SMILING [2]).
> - When considering other adversarial techniques, we chose AMP as a baseline because it is an improved formulation of previous state-only methods like GAiFO [4] and is the strongest state-of-the-art baseline in this field (it is still a prominent algorithm used for learning such complex policies [5]).

---

> > ### Author Response · Authors · 2024-11-16
> > **References used in our response**
> >
> > [1] Wang B, Wu G, Pang T, Zhang Y, Yin Y. DiffAIL: Diffusion Adversarial Imitation Learning. InProceedings of the AAAI Conference on Artificial Intelligence 2024 Mar 24 (Vol. 38, No. 14, pp. 15447-15455).
> >
> > [2] R. Wu, Y. Chen, G. Swamy, K. Brantley, and W. Sun, “Diffusing States and Matching Scores: A New Framework for Imitation Learning,” Oct. 17, 2024, arXiv: arXiv:2410.13855. Accessed: Oct. 19, 2024. [Online]. Available: http://arxiv.org/abs/2410.138
> >
> > [3] Ho J, Ermon S. Generative adversarial imitation learning. Advances in neural information processing systems. 2016;29.
> >
> > [4] Torabi F, Warnell G, Stone P. Generative adversarial imitation from observation. arXiv preprint arXiv:1807.06158. 2018 Jul 17.
> >
> > [5] L'Erario G, Hanover D, Romero A, Song Y, Nava G, Viceconte PM, Pucci D, Scaramuzza D. Learning to Walk and Fly with Adversarial Motion Priors. arXiv preprint arXiv:2309.12784. 2023 Sep 22.

---

> > > ### Comment · Reviewer_hjSn · 2024-11-17
> > > **Thanks for the response**
> > >
> > > I appreciate the author's effort in clarifying the relationship with existing diffusion-based LfO work. DiffAIL still uses adversarial training, and since SMILING is not published, it is not fair to ask the author to compare their method with it. Overall, my concerns about the originality of NEAR have been addressed.

---

### Official Review · Reviewer_Tb6J · 2024-11-02

**Soundness:** 4
**Presentation:** 4
**Contribution:** 2
**Rating:** 6
**Confidence:** 4

**Summary:**

This paper proposes a method to perform imitation learning in absence of the expert's actions (i.e., only having access to the state trajectories). The proposed algorithm, NEAR, uses noise-conditioned score networks to model the probability distribution of the trajectories in the dataset. In this way, NEAR obtains a reward signal for imitation learning that does not depend on an adversarial network, as it is the case in current state of the art methods like AMP. In this way, known issues with learning in an adversarial setting are avoided. NEAR successfully learns to imitate reference trajectories, with similar performance to AMP and with smooth motion.

**Strengths:**

The paper is very well written. The algorithm presented in this paper (NEAR) is explained in detail. The authors clearly explain how this work is positioned within the field of motion imitation, providing helpful context information about adversarial imitation learning and noise-conditioned score networks.

**Weaknesses:**

It is unclear how Figure 1 has been generated. It is used for illustration purpose, however some more information about the energy function and the adversarial reward are necessary. I also do not understand the choice of using different scales for the two rewards, and why the energy is high around the agent's trajectory while the advesarial reward is low. I would ask the authors to detail how the energy function and the adversarial reward have been learnt (also in the supplementary material if it does not fit the main text).

The comparison with adversarial imitation learning methods is clear and well done. However, another possibility for imitation learning would  be, e.g., to just provide a reward signal in the form of the L2 distance between the states visited by the policy and the ones to imitate. NEAR seems, in fact, a more sophisticated version of this simple method. It would be important to explain why other non-adversarial imitation learning methods are excluded from the evaluation, and their differences and similarities with NEAR.

The algorithm presented in this paper does not seem to be a clear improvement over the baseline method (AMP). While the evaluation metric "spectral arc length" seems to favor NEAR, motions generated with AMP are generally closer to the ground truth. The authors should motivate why NEAR is better or at least a good alternative to AMP, e.g., proving lower sensitivity to the hyperparamenter choice, faster learning, less variance, more sample efficiency ... I encourage the authors to also compare the learning curves of the two methods, possibly including the number of interactions with the environment and the wall time necessary for convergence.

**Questions:**

* The authors mention that they use 20 test episodes to obtain the average performance, which sounds low compared to other RL papers. How large is the variability in performance across episodes? The confidence levels are provided across random seeds, so the variability of the performance within a seed is not evident.

* You mention that AMP is less affected by data availability than NEAR. Shouldn't it also be a problem for AMP when data is scarce, since the task of the discriminator might become too easy when it can perfectly remember all the ground truth trajectories?

* I did not fully understand why the energy function works well as a reward signal. If the energy is high when a sample is likely to be generated by the probability distribution of the ground truth data, why does the policy follow a trajectory instead of just reaching a high probability state? I thought that one reason can be that the energy function depends on the current state, so it will assign high energy only to the states that, according to the dataset, follow the current one with high probability. While this concepts are likely trivial for the authors, they should be more clearly explained in the paper for the less familiar reader to fully understand why the algorithm works. I would propose to use the example from Figure 1 to qualitatively describe why the displayed energy function is a good reward signal, if the energy changes as the agent moves, and other high-level considerations.

I commit to increase the score if my questions and doubts highlighted in the "weaknesses" section are carefully addressed.

---

> ### Author Response · Authors · 2024-11-16
> **Review responses & paper revisions**
>
> Dear reviewer, thank you for your feedback, we appreciate the positive comments and the clearly explained questions/limitations. We have now uploaded a revised paper and supplementary material. Please find clarifications and paper changes below (revisions based on your feedback are coloured red). We hope these sufficiently establish the contribution of the paper.
>
>
> >It is unclear how Figure 1 has been generated. It is used for illustration purposes, however, some more information about the energy function and the adversarial reward are necessary. I also do not understand the choice of using different scales for the two rewards, and why the energy is high around the agent's trajectory while the adversarial reward is low. I would ask the authors to detail how the energy function and the adversarial reward have been learnt (also in the supplementary material if it does not fit the main text).
> - **Figure 1 Clarity:**
>     - Thank you for pointing out the clarity issues with Figure 1. This was also pointed out by another reviewer and we have taken several steps to ensure that the figures are clear to the reader.
>     - We have now added an additional explanation of the maze domain in the Figure 1 caption. We have also added another appendix section (Appendix B.3 MAZE DOMAIN DETAILS) detailing the procedures to generate Figure 1.
>     - We explain these briefly here. The energy-based reward function in Figure 1 was learnt using the energy-based NCSN model described in this paper. We trained the model with $\sigma=20.0$ on expert trajectories in the maze domain. The adversarial reward function was obtained by training AMP in the maze domain and saving the checkpoints at a comparable point in training.
>     - We request the reviewer to kindly provide feedback on the revised figure clarity.
> - **Scaling Differences:**
>     - The scale difference between the energy-based and adversarial rewards is a result of the way in which the energy-based model and discriminator are defined.
>     - In AMP, the discriminator is a least-squares regression classifier (please refer to Section 5.2 in [1]) and returns a value between [-1,1]. We chose to keep the same scale in the illustration to maintain correctness with the original AMP paper.
>     - The energy function in score-based models is typically un-normalised. However, we chose to illustrate it on a scale of [0,1] to avoid unnecessary complications in the illustration.
>     - We agree that this results in a somewhat confusing comparison between the two reward functions. We are happy to remove the scaling details altogether as they are not integral to the figure’s meaning.
> - **Energy/Discriminator Value Clarifications:**
>     - Typically, low energies indicate closeness to the data distribution. However, for simpler downstream RL, we chose to invert the sign of the learnt energy function such that maximising the energy reward means closeness to the expert (please refer to 4.1 LEARNING ENERGY FUNCTIONS)
>     - The adversarial reward around the data distribution is low because of the fluctuating nature of the discriminator's decision boundary. In this case, $p_G$ has a high intersection with $p_D$. The decision boundary is shown such that the discriminator prediction is low (close to -1) near $p_G$ and gradually changes to be higher towards the remaining (non-intersecting) portion of $p_D$. In our experiments, we recorded highly fluctuating adversarial rewards for AMP. We have now also included a new appendix section (Appendix E) providing additional empirical evidence of the instability and non-smoothness in AMP. We hope this inclusion clarifies the ideas behind Figure 1.
>
> >It would be important to explain why other non-adversarial imitation learning methods are excluded from the evaluation and their differences and similarities with NEAR.
> - To the best of our knowledge, NEAR is novel in its use of score-based models and energy-based rewards for imitation learning from observation. Other non-adversarial techniques for imitation learning (like [4] and its variants) primarily operate in a behaviour-cloning fashion. They hence require the actions executed by the expert. In contrast, NEAR operates in a state-only fashion.
>
> >While the evaluation metric "spectral arc length" seems to favor NEAR, motions generated with AMP are generally closer to the ground truth.
> - We disagree with the claim that motions generated by AMP are closer to the ground truth. We believe that NEAR is closer to the ground truth in most cases. The provided supplementary videos show qualitatively that our approach generates closer-to-ground truth motions. We have now also included additional videos of the ground truth motions. Please, indicate to us if the supplementary videos helped.

---

> > ### Author Response · Authors · 2024-11-16
> > **Review responses continuation**
> >
> > >However, another possibility for imitation learning would be, e.g., to just provide a reward signal in the form of the L2 distance between the states visited by the policy and the ones to imitate. NEAR seems, in fact, a more sophisticated version of this simple method.
> > - Yes, it is indeed possible to use the L2 distance between the states in the expert demonstrations and the states visited by the policy. This has been explored in previous works like DeepMimic [3]. However, there are several practical reasons for learning an energy function instead of just using the L2 distance. First, it is not unclear as to which reference state the L2 norm must be computed with. Any given state in the policy-generated motion can be mapped to a wide range of expert states. Mapping to a few specific states is only possible if the goal is to replicate a specific expert trajectory. This already limits the generalisation capabilities of the algorithm.
> > - An alternative approach could be to compute the average L2 distance to the whole expert demonstration manifold. This is also what the energy function implicitly captures. Doing this manually during RL would lead to practical issues and slow computation. Further, sampling from the dataset at every step might lead to unstable reward functions as the target of the L2 distance would be changing every time.
> > - Having said that,  we agree that it is important to compare again such simple baselines. We also plan to add this as a baseline in our future work on NEAR.
> >
> > >The authors should motivate why NEAR is better or at least a good alternative to AMP, e.g., proving lower sensitivity to the hyperparameter choice, faster learning, less variance, more sample efficiency ... I encourage the authors to also compare the learning curves of the two methods, possibly including the number of interactions with the environment and the wall time necessary for convergence.
> > - Thank you for this suggestion. While the motive for this paper was to introduce a new idea for energy-based reward learning, we also realise that a quantitative comparison of benefits is necessary to establish our contribution. We have now included the following additional details to better compare NEAR and AMP.
> >     - We have included additional experiments (in Appendix E CHALLENGES OF ADVERSARIAL IL & EXPERIMENTS ON THE AMP DISCRIMINATOR ) on the AMP discriminator to highlight its issues like high reward function variance, and non stationarity. These experiments highlight the core problems with adversarial IL that our paper hopes to address.
> >     - We have included Table 5 which shows the number of training iterations and environment interactions of both algorithms. Both AMP and NEAR are ultimately trained for the same number of RL interactions. All results in this paper compare both algorithms at the same point in training.
> >     - We have included Table 7 which shows the wall time of both algorithms. While NEAR has a slightly larger overall wall time (considering NCSN), it has a lower RL wall time than AMP. We believe that while the wall time of NEAR is slightly higher, it is not a large difference to discount the use of NEAR as an alternative to AMP. We would also like to point out that our research mainly aims to show better and more stable alternatives to adversarial reward learning.
> >     - Finally, we also plan to include additional experiments on the hyperparameter sensitivity of AMP. However, given the tight time window of rebuttals, we are not certain that we can conduct these experiments in time. We assure the reviewer that these will be included in the arxiv submission.
> >
> > >You mention that AMP is less affected by data availability than NEAR. Shouldn't it also be a problem for AMP when data is scarce, since the task of the discriminator might become too easy when it can perfectly remember all the ground truth trajectories?
> > - Yes, it is true that a perfect discriminator (one that can perfectly distinguish between $p_D$ and $p_G$) is indeed an issue for AMP. This is because a perfect discriminator would nearly always provide zero rewards to the agent (unless it exactly reproduces the demonstration data).
> > - However, we believe that at the scale of the datasets used in this paper, data unavailability does not pose any realistic problems to AMP. A perfect discriminator can be achieved just as easily with a fairly large amount of data (please refer to [2]). Hence, AMP is prone to the same issues of perfect discrimination even with multi-clip imitation data. This is why we state that AMP performs more or less the same under single-clip settings.

---

> > > ### Author Response · Authors · 2024-11-16
> > > **Review responses continuation**
> > >
> > > >The authors mention that they use 20 test episodes to obtain the average performance, which sounds low compared to other RL papers. How large is the variability in performance across episodes? The confidence levels are provided across random seeds, so the variability of the performance within a seed is not evident.
> > > - In this paper, performance metrics are computed by rolling out the learnt policies deterministically across several (4096) independent parallel environments. Each individual environment is initialized at some random point in the expert's motion trajectories (please refer to Appendix B.2.3 EVALUATION METRICS).
> > > - We find that the performance across independent episodes with the same initialisation is nearly identical. However, the performance varies depending on initialisation. This can greatly bias the results against either algorithm. For example, performance measured only from the initial point in the expert's motions couldn't provide any insights on imitation capabilities from some intermediate state.
> > > - Hence, we record the average performance of the top 20 rollouts (across several randomly initialized episodes). This gives an overall indication of the "best" performance of a learnt policy regardless of initialisation. That being said, towards the end of training, the performance across multiple random initial states is also nearly identical for both algorithms.
> > >
> > > >I did not fully understand why the energy function works well as a reward signal. If the energy is high when a sample is likely to be generated by the probability distribution of the ground truth data, why does the policy follow a trajectory instead of just reaching a high probability state? I thought that one reason can be that the energy function depends on the current state, so it will assign high energy only to the states that, according to the dataset, follow the current one with high probability. While this concepts are likely trivial for the authors, they should be more clearly explained in the paper for the less familiar reader to fully understand why the algorithm works. I would propose to use the example from Figure 1 to qualitatively describe why the displayed energy function is a good reward signal, if the energy changes as the agent moves, and other high-level considerations.
> > > - Thank you for this insight. We would like to clarify that in our algorithm, we learn an energy function over the expert's state transitions. Hence, the policy is highly rewarded for producing state transitions that are similar to the experts' (the same is also done in AMP). Indeed, if the energy function was learnt over individual states (as opposed to state transitions), then the policy would probably just reach a high energy state and stop.
> > > - While we mention the use of state transitions at several points in the paper, we have now clarified this explicitly on line 351.
> > > - In the maze domain, the actual reward functions are in the 4-dimensional space of state transitions. In Figure 1, we show these 4D reward functions as 2D representations. To obtain this 2D representation, we compute the average reward in the agent’s reachable set at every point in a discretised grid on the maze domain. We have added an additional explanation of the procedures to generate the figures (Appendix B.3 MAZE DOMAIN DETAILS). We have also mentioned this 2D projection in the caption of Figure 1 (line 125) and have pointed to the appendix section. We hope this sufficiently clarifies the idea of using state transitions in our paper.
> > >
> > > [1] Peng XB, Ma Z, Abbeel P, Levine S, Kanazawa A. Amp: Adversarial motion priors for stylized physics-based character control. ACM Transactions on Graphics (ToG). 2021 Jul 19;40(4):1-20.
> > >
> > > [2] Arjovsky M, Bottou L. Towards principled methods for training generative adversarial networks. arXiv preprint arXiv:1701.04862. 2017 Jan 17.
> > >
> > > [3] Peng XB, Abbeel P, Levine S, Van de Panne M. Deepmimic: Example-guided deep reinforcement learning of physics-based character skills. ACM Transactions On Graphics (TOG). 2018 Jul 30;37(4):1-4.
> > >
> > > [4] Chi C, Xu Z, Feng S, Cousineau E, Du Y, Burchfiel B, Tedrake R, Song S. Diffusion policy: Visuomotor policy learning via action diffusion. The International Journal of Robotics Research. 2023:02783649241273668.

---

> > > ### Comment · Reviewer_Tb6J · 2024-11-17
> > >
> > > *To the best of our knowledge, NEAR is novel in its use of score-based models and energy-based rewards for imitation learning from observation. Other non-adversarial techniques for imitation learning (like [4] and its variants) primarily operate in a behaviour-cloning fashion. They hence require the actions executed by the expert. In contrast, NEAR operates in a state-only fashion.*
> > >
> > > When actions are not part of the dataset, one can still train the agent to track the state trajectories in the dataset (see, e.g., [1]). You write:
> > >
> > > *First, it is not unclear as to which reference state the L2 norm must be computed with.*
> > >
> > > Here the task would be to track expert trajectories, so you would sample a trajectory and just provide a reward proportional to $||s_t - s^*_t||^2$.
> > >
> > > *Any given state in the policy-generated motion can be mapped to a wide range of expert states.*
> > >
> > > Indeed you would have to sample a trajectory per episode. These other approach you propose
> > >
> > > *An alternative approach could be to compute the average L2 distance to the whole expert demonstration manifold*
> > >
> > > seems indeed impractical. However, I only partially agree with
> > >
> > > *This already limits the generalisation capabilities of the algorithm.*
> > >
> > > I can imagine that using an energy function is better when the expert trajectories are not too many, but if the dataset is large enough, direct imitation of expert trajectories leads to (maybe surprisingly) very good generalization to unseen trajectories (I refer again to [1] but there are surely other papers showing this). So it would have been a valuable addition showing when replacing direct expert imitation with an energy-based reward is better and when it is worse.
> > >
> > > [1] Luo, Zhengyi, et al. "Perpetual humanoid control for real-time simulated avatars." Proceedings of the IEEE/CVF International Conference on Computer Vision. 2023.

---

> > > > ### Comment · Reviewer_Tb6J · 2024-11-17
> > > > **NEAR better than AMP in most cases**
> > > >
> > > > *We disagree with the claim that motions generated by AMP are closer to the ground truth. We believe that NEAR is closer to the ground truth in most cases. The provided supplementary videos show qualitatively that our approach generates closer-to-ground truth motions. We have now also included additional videos of the ground truth motions. Please, indicate to us if the supplementary videos helped.*
> > > >
> > > > I do not understand this comment as AMP has lower pose error in all but one task (and in one it's the same as NEAR), according to table 1. Also in the text you write:
> > > >
> > > > *In most experiments, NEAR is closer to the expert in terms of the spectral arc length while AMP has a better pose error.*
> > > >
> > > > These metrics are more reliable than qualitative inspection of the videos, where small differences in pose error cannot be detected.

---

> > > > > ### Author Response · Authors · 2024-11-18
> > > > >
> > > > > >I do not understand this comment as AMP has lower pose error in all but one task (and in one it's the same as NEAR), according to table 1. These metrics are more reliable than qualitative inspection of the videos, where small differences in pose error cannot be detected.
> > > > > - We agree that AMP is indeed usually quantitatively better in terms of pose error. However, NEAR performs better in terms of spectral arc length. We think that qualitative results are also a valid complement to these quantitative ones and that perhaps the metrics don't perfectly capture the smaller nuances of “natural” imitation. We chose these metrics because of their use in prior works like AMP. In any case, to address your original comment:
> > > > > > The authors should motivate why NEAR is better or at least a good alternative to AMP…
> > > > > - We have now provided additional results (on the challenges of AMP) that we explained in one of our previous answers. We are also planning to add a few more experiments on hyperparameter sensitivity and perfect discrimination (under varying dataset sizes).

---

> > > ### Comment · Reviewer_Tb6J · 2024-11-18
> > > **Data availability AMP vs NEAR**
> > >
> > > With my comment I was referring to this paragraph:
> > >
> > > *Finally we notice that NEAR performs poorly in single-clip imitation tasks, highlighting the challenges of accurately capturing the expert’s data distribution in data-limited conditions. Conversely, AMP is less affected by data unavailability since the discriminator in AMP is simply a classifier and does not explicitly capture the expert’s distribution.*
> > >
> > > With your further clarification:
> > >
> > > *However, we believe that at the scale of the datasets used in this paper, data unavailability does not pose any realistic problems to AMP. A perfect discriminator can be achieved just as easily with a fairly large amount of data (please refer to [2]). Hence, AMP is prone to the same issues of perfect discrimination even with multi-clip imitation data. This is why we state that AMP performs more or less the same under single-clip settings.*
> > >
> > > you seem to suggest that AMP can learn a perfect discriminator both with a single clip and with the full dataset. But you also write:
> > >
> > > " a perfect discriminator would nearly always provide zero rewards to the agent (unless it exactly reproduces the demonstration data)."
> > >
> > > As it is unlikely that the agent reproduces the expert data exactly, why do you think AMP works? While the conclusion of your experiments is that AMP works more or less fine both with single clip and for the full dataset, the conclusion I would draw from your arguments is that AMP should not work in either case, as a perfect discriminator is a critical problem for the algorithm.

---

> > > > ### Author Response · Authors · 2024-11-25
> > > > **Additional empirical results on perfect discrimination**
> > > >
> > > > Dear reviewer, apologies for the delay in responding to this comment. We took some time to analyse this theoretically and empirically.
> > > >
> > > > > You seem to suggest that AMP can learn a perfect discriminator both with a single clip and with the full dataset
> > > > - Yes, we believe that given the relatively small datasets used in our paper, AMP can indeed learn a perfect discriminator even in multi-clip settings. This has been analysed in rigorous theoretical detail in [2].
> > > > - We have now replicated the perfect discriminator experiments from [2] (refer to Figure 1 in [2]) on AMP. We observe nearly the same results/patterns as described in [2]. These results are available in Appendix E.1.3 in the revised paper (with theoretical reasoning in lines 1052-1069). A summary of these experiments is below.
> > > >     - We train AMP on the walking dataset (largest motion dataset in this paper) for varying training times. In each trial, policy training was paused after a certain cut-off point (say 5 e6 RL interactions) and a discriminator was retrained to distinguish between $p_G$ and $p_D$. We observe that in all cases, the discriminator very quickly learns to perfectly distinguish between the two distributions. This means that $supp(p_G)$ and $supp(p_D)$ might be non-continuous and disjoint.
> > > >     - With a perfect discriminator, the rewards received by the agent also rapidly become either zero or constant (depending on how the classifier predictions are used as a reward function). This would lead to poor RL.
> > > >
> > > > > As it is unlikely that the agent reproduces the expert data exactly, why do you think AMP works?
> > > > - Our experiments show that the perfect discrimination issue indeed exists for multi-clip settings. However, as rightly pointed out by the reviewer, it does not really explain why AMP works in the first place.
> > > > - Unfortunately, at the moment we are not sure about this. Why standard GANs work despite perfect discrimination also seems to be unanswered in [2]. This is indeed a very interesting research direction that must be further explored via a dedicated analysis. We believe that for the purposes of our current paper, showing empirical evidence of perfect discrimination strengthens the argument that energy-based rewards are better than adversarial ones (i.e. even in the best case for AMP, perfect discrimination is still not nice to have).
> > > >
> > > > We hope this analysis clarifies your concerns and are happy to explain anything in further detail.
> > > >
> > > >
> > > >
> > > > [2] Arjovsky M, Bottou L. Towards principled methods for training generative adversarial networks. arXiv preprint arXiv:1701.04862. 2017 Jan 17.

---

> > > > > ### Comment · Reviewer_Tb6J · 2024-11-26
> > > > >
> > > > > Thank you for the further experiments. I have briefly checked the AMP paper and they use a probability measure to decide whether the transition (s,s') is part of the dataset. Therefore, even if almost 0 for all generated transitions, there might still be a relative ranking (more similar to the dataset -> higher probability) which makes the whole process work. I agree that if the discriminator outputs exactly 0 for all generated transitions then AMP will stop working, but it seems not to be the case in practice.

---

> > ### Comment · Reviewer_Tb6J · 2024-11-17
> > **Additional clarifications**
> >
> > I quickly respond to the first comment so that you have time to further refine the first figure before the end of the discussion period.
> >
> > I think the main source of confusion about figure 1 is the fact that the energy function is not just a function of the 2D position in the maze, but of the state transition (as you correctly point out in another answer). As you understood from my doubts about whether the energy function is a good reward signal, I was mislead by this figure, which shows the energy as a function of the position in the maze. But it should also be a function of the position of the ant (please correct me if I am wrong, but this time my understanding should be correct). Then what is the sense of showing the energy function when the agent is in the top-left corner, and the adversarial reward when the agent is in the bottom-left corner? The two functions are hardly comparable. I think it would be much more helpful to show, e.g., how the energy function and the adversarial reward change with the position of the agent (stacking some frames one after the other), also highlighting where the goal is (the goal should also be part of the state, and therefore influence the shape of the function). Neglecting these important variables when visualizing the energy and the adversarial reward, in my opinion, does not help clarifying their features. Most importantly, the comparison must be in equal conditions (agents at the same place, same goal, ...), as there are many variables to think about and the visualization should focus on a single one (in this case: left->energy, right->adversarial).

---

> > > ### Author Response · Authors · 2024-11-18
> > > **Updated figure 1 & requesting feedback**
> > >
> > > Dear Reviewer, we appreciate the quick response! We agree that the way we illustrated it previously did indeed lead to misunderstandings and confusion for the reader. We thank you for pointing this out and for helping us improve the illustrations. The revised illustration compares both reward functions under the same conditions. We request you to please provide feedback on the updated figure.
> > >
> > > >I think the main source of confusion about figure 1 is the fact that the energy function is not just a function of the 2D position in the maze, but of the state transition (as you correctly point out in another answer). As you understood from my doubts about whether the energy function is a good reward signal, I was mislead by this figure, which shows the energy as a function of the position in the maze. But it should also be a function of the position of the ant (please correct me if I am wrong, but this time my understanding should be correct).
> > > - Apologies for the confusion. To clarify, in NEAR, the energy function is constant throughout training. This means that $r(s’ | s)$ (we use $r()$ to mean reward function) is stationary for any state $s$. It does not depend on the distribution of policy visited states($p_G$) and is illustrated with different $p_G$'s to highlight this point.
> > > - In AMP, because of the discriminator's min-max nature, the reward is also indirectly dependent on the distribution of policy visited states (2D positions in the maze domain).
> > > - We have updated Figure 1 to clarify this point. Now, figure 1 shows the energy-based and adversarial rewards evaluated around a single zoomed-in state at various points in training (thanks for the suggestion). The energy-based reward stays the same because it is not updated after pre-training the energy function. The adversarial reward keeps changing throughout training (meaning that $r(s’ | s)$ keeps changing through time even for a fixed $s$).
> > > - In the new figure, we also illustrate the non-smooth nature of the adversarial reward and contrast it with the smooth nature of the energy-based one. To do this, we maintain parts of the previous figure and plot the sum over all rewards in the agent’s reachable set at every state in the domain. This is meant to compare the rewards globally. Even in this case, everything else about the two functions remains the same (i.e they are evaluated at every state and illustrated without showing the agent’s position).
> > >
> > > >Then what is the sense of showing the energy function when the agent is in the top-left corner, and the adversarial reward when the agent is in the bottom-left corner?
> > > - As clarified above, the agent’s position is irrelevant to the energy-based reward. However, we understand that illustrating the agent in the energy function plot would lead to confusion. We have now removed the agent from both plots and have instead highlighted the fact that the energy-based reward does not depend on $p_G$.
> > >
> > > >also highlighting where the goal is (the goal should also be part of the state, and therefore influence the shape of the function).
> > > - In this task, the goal was not part of the agent’s state. Again, it was added to the illustration only for the benefit of the reader. However, we understand how it would be misleading. We have also removed the goal from the plots in the illustration.
> > > - In general, the maze domain could be converted to a goal-conditioned RL problem (as we did in some main experiments on humanoid goal reaching). However, we believe that adding the goal feature to the states in the maze domain does not provide additional comparisons between the two reward function
> > >
> > > Hope the revised figure is easier to understand. We are of course happy to made any other suggested modifications!

---

> ### Comment · Reviewer_Tb6J · 2024-11-19
>
> I apologize if I sound pedantic, but I still have a few doubts after your last answer.
>
> First of all, thanks for updating the figure, I find that the new one nicely shows an important property of your algorithm and a benefit of removing the adversarial component from the training process. However, this was not really what I meant with my comment, but I am still happy it lead to an improvement.
>
> My misunderstanding was not about r(s, s') changing throughout the training, I understand that the function is learnt from a static dataset and not updated. My question is simpler than that. I don't know the details about the state, but I believe it includes the position of the agent in the 2D maze (or some representation of it). So I believe that to plot the energy function you fixed s and plotted r(s,s') where in s' you modify the 2D position in the maze. Please correct me if I am wrong.
>
> Now, s depends on the current position of the agent in the maze. In the previous figure, you plotted r(s, s') for the energy function, but you used a different s for the adversarial reward (the agent was in a different location). That's why I said it would be more useful if you plotted r: s' -> r(s,s') keeping the same s, but using first r=energy and then r=adversarial reward. And then I proposed that you could take s within a trajectory and show how the "heatmap" of  r: s' -> r(s,s') changes as a function of s (you could stack frames one after the other). I assume it would show some interpretable pattern, e.g., that the states behind the agent become lower reward states as it moves, as the reward should promote the agent to go forward and not backward. I believe this would immediately convince the reader that the energy of a transition is a good reward function.

---

> > ### Author Response · Authors · 2024-11-19
> >
> > Dear reviewer, thanks again for the fast response. We appreciate the continuous feedback.
> >
> > >  I don't know the details about the state, but I believe it includes the position of the agent in the 2D maze
> > - Yes, the agent's state is a vector of its 2D coordinates in the maze.
> >
> > > I believe that to plot the energy function you fixed s and plotted r(s,s') where in s' you modify the 2D position in the maze
> > - Yes, you are correct. We fixed $s$ (green circle in the figure) and plotted $r(s'|s)$ in a small window around s. This is meant to show the possible reward the agent would get when transitioning to another state $s'$ from $s$.
> >
> > > In the previous figure, you plotted r(s, s') for the energy function, but you used a different s for the adversarial reward (the agent was in a different location)
> > - This might be the reason for the misunderstanding. In the previous figure (which is now the subfigure on the right side), we plot $\sum_{\text{all } s' \text{reachable from } s} r(s'|s)$. This means that it represents the overall reward landscape and not a reward from any single state. In the previous illustration, we confusingly showed different agent locations for the two rewards, although both were computed for all states in the domain. We have now fixed this by removing the states in the right subfigure.
> > - To summarise, in the new figure, each frame of the middle plot shows $r(s'|s)$ for a **fixed** state for both reward functions (we believe that this is what you requested in your comment). You can notice that the energy reward is higher for states below the green state, meaning that the agent is positively rewarded for going forward.
> >
> > > I proposed that you could take s within a trajectory and show how the "heatmap" of r: s' -> r(s,s') changes as a function of s (you could stack frames one after the other)
> > - Our motivation for Figure 1 was to (i) illustrate that the adversarial reward is non-stationary and (ii) illustrate that the adversarial reward is non-smooth. We believe that showing heatmaps of the reward functions for different states in a trajectory would indeed intuitively show how the reward motivates the agent to move forward. However, showing this would not help with highlighting these two points.
> > - Instead, we are happy to add this (illustration of rewards for different states in a trajectory) to the appendix. We will also tweak the energy-based reward in the current figure to better highlight that it rewards $s'$ states in front of the agent higher than ones behind the agent. We will include these revisions in the next few days.
> >
> > We hope you understand our motivations behind Figure 1. We are happy to clarify any questions further.

---

> > > ### Comment · Reviewer_Tb6J · 2024-11-19
> > > **Thank you for the clarifications**
> > >
> > > I want to thank the authors for the detailed responses, I am now more confident in recommending acceptance of the paper.

---

> > > > ### Author Response · Authors · 2024-11-19
> > > >
> > > > Dear reviewer, thank you for taking the time to discuss this with us. We're glad we could help clarify things.
> > > >
> > > > **Upcoming changes**
> > > > - Minor tweaks to Figure 1 to highlight that the energy function rewards states in front of the agent higher than ones behind the agent
> > > > - We have some theories for why AMP is less affected by data unavailability and why the perfect discriminator problem does not just result in completely bad performance. We are conducting some experiments on perfect discrimination at the moment and will try to answer this via an added section in the appendix.
> > > > - We plan to also conduct some hyperparameter sensitivity analysis on AMP and NEAR. This will be added to the paper if we can obtain results by the rebuttal deadline. In the future, we would also conduct experiments with other baselines (like the L2 distance reward idea you suggested).
> > > >
> > > > Thanks again.

---

### Official Review · Reviewer_PK39 · 2024-11-03

**Soundness:** 2
**Presentation:** 2
**Contribution:** 2
**Rating:** 6
**Confidence:** 4

**Summary:**

This paper introduces Noise-Conditioned Energy-based Annealed Rewards (NEAR), a novel framework for imitation learning from observation using energy-based generative models. NEAR leverages denoising score matching to learn smooth representations of the expert's motion distribution and uses these energy functions as rewards. Unlike adversarial imitation learning approaches, NEAR avoids unstable min-max optimization, achieving smoother and more stable reward signals. Additionally, an annealing strategy progressively transitions between energy functions to provide more refined guidance for the agent’s policy. NEAR is evaluated on complex humanoid tasks, showing promising results when compared to state-only adversarial imitation learning baselines like Adversarial Motion Priors (AMP) in terms of motion quality, stability, and imitation accuracy.

**Strengths:**

1. NEAR performs well on a range of complex motion tasks, including stylized walking, running, and martial arts. The results demonstrate competitive imitation accuracy and smoothness compared to AMP, particularly in complex tasks where AMP struggles with stability.
2. The paper includes ablation studies to explore the impact of key components.

**Weaknesses:**

1. NEAR’s effectiveness is primarily evaluated in humanoid tasks, which are continuous and physics-driven. The framework’s applicability in other types of imitation learning tasks, especially those with discrete actions or diverse goal-oriented, is not fully explored.
2. NEAR requires training a noise-conditioned energy model, which can be computationally intensive. A detailed comparison of training costs relative to other methods, particularly in terms of time and resources, would be beneficial.
3. NEAR is only compared against one baseline - AMP and it doesn't seem to always be the winner (but has higher variance in most cases) despite the additional complexity of learning an energy network.

**Questions:**

Could you provide a comparison of NEAR’s computational requirements (e.g., training time, GPU hours) relative to other baselines like AMP?

---

> ### Author Response · Authors · 2024-11-16
> **Review responses & paper revision**
>
> Dear reviewer, thank you for your feedback. We have now submitted a revised paper. Please find clarifications and paper changes below. The text in some parts is coloured to indicate which revisions are associated with which reviewer (this will be reverted after rebuttals).  We hope these clarifications help explain the contribution of our paper and request you to reconsider your score.
>
> >NEAR requires training a noise-conditioned energy model, which can be computationally intensive. A detailed comparison of training costs relative to other methods, particularly in terms of time and resources, would be beneficial.
> - Thank you for highlighting this. In their typical use case (very high dimensional image/video generation) score-based generative models are indeed computationally intensive. However, we believe that the typical use case of score-based models is not comparable to their use in NEAR (relatively low dimensional state-transition pairs). We train the energy-based reward function for less than 150k training iterations in all cases. This is more than sufficient to learn the energy-based reward functions that are used in NEAR.
> - We have now included two additional tables providing information on the training costs. Table 5 in Appendix B.2.4 provides more information on the training iterations and environment interactions of both NEAR and AMP in our experiments. Both algorithms' RL policies are trained for the same number of environment interactions. Table 7 in Appendix C compares the wall-clock training times of NEAR and AMP. We find that overall NEAR requires slightly more training time than AMP. Across all tasks, NCSN contributes to less than 30% of the total computational time of NEAR, with RL accounting for the majority of the computation load and wall-clock time. Interestingly, AMP always has a larger reinforcement learning computational load. This is expected as AMP is learning both the policy and the reward function simultaneously.
>
> >Could you provide a comparison of NEAR’s computational requirements (e.g., training time, GPU hours) relative to other baselines like AMP?
> - Please see the previous answer and the newly included Table 5 and Table 7.
>
> >NEAR is only compared against one baseline - AMP and it doesn't seem to always be the winner (but has higher variance in most cases) despite the additional complexity of learning an energy network.
> - To the best of our knowledge, AMP is the only comparable baseline for imitation learning from observation in such high-dimensional physics-based problems. It is an improved formulation of previous state-only methods like GAiFO [1] and is the strongest state-of-the-art baseline in this field (it is still a prominent algorithm used for learning such complex policies [2]).
> - We agree that NEAR does have a slightly higher overall computational load than AMP (please also see the answer above and Tables 5 and 7). However, we would also like to point out that our research mainly aims to provide a better and more stable alternative to adversarial imitation learning. To that end, we believe that NEAR has a better mechanism for learning reward functions that is not prone to instability, non-stationarity and learning non-smooth rewards. While NEAR does have more variance in some tasks, this is mainly due to the limitations of annealing. We believe that the underlying reward learning mechanism of NEAR is still superior to that of AMP.
> - To better highlight these limitations of AMP, we have now added another appendix section with additional explanations and empirical analysis (Appendix E) on the AMP discriminator. We hope that this additional information helps situate NEAR as a better alternative for reward learning in imitation learning from observation problems.
>
> >NEAR’s effectiveness is primarily evaluated in humanoid tasks, which are continuous and physics-driven. The framework’s applicability in other types of imitation learning tasks, especially those with discrete actions or diverse goal-oriented, is not fully explored.
> - We believe that the scope of our paper aligns well with the experimental analyses from related works like [1,3,4,5,6,7,8]. All these prior works in the field of generative imitation learning also only conduct experiments in continuous domains. We believe that NEAR is situated amongst these works and our experiments are comparable to these related works.

---

> > ### Author Response · Authors · 2024-11-16
> > **References used in our response**
> >
> > [1] Torabi F, Warnell G, Stone P. Generative adversarial imitation from observation. arXiv preprint arXiv:1807.06158. 2018 Jul 17.
> >
> > [2] L'Erario G, Hanover D, Romero A, Song Y, Nava G, Viceconte PM, Pucci D, Scaramuzza D. Learning to Walk and Fly with Adversarial Motion Priors. arXiv preprint arXiv:2309.12784. 2023 Sep 22.
> >
> > [3] Peng XB, Abbeel P, Levine S, Van de Panne M. Deepmimic: Example-guided deep reinforcement learning of physics-based character skills. ACM Transactions On Graphics (TOG). 2018 Jul 30;37(4):1-4.
> >
> > [4] Ho J, Ermon S. Generative adversarial imitation learning. Advances in neural information processing systems. 2016;29.
> >
> > [5] Peng XB, Ma Z, Abbeel P, Levine S, Kanazawa A. Amp: Adversarial motion priors for stylized physics-based character control. ACM Transactions on Graphics (ToG). 2021 Jul 19;40(4):1-20.
> >
> > [6] Tessler C, Kasten Y, Guo Y, Mannor S, Chechik G, Peng XB. Calm: Conditional adversarial latent models for directable virtual characters. InACM SIGGRAPH 2023 Conference Proceedings 2023 Jul 23 (pp. 1-9).
> >
> > [7] Prasad A, Lin K, Wu J, Zhou L, Bohg J. Consistency policy: Accelerated visuomotor policies via consistency distillation. arXiv preprint arXiv:2405.07503. 2024 May 13.
> >
> > [8] Chi C, Xu Z, Feng S, Cousineau E, Du Y, Burchfiel B, Tedrake R, Song S. Diffusion policy: Visuomotor policy learning via action diffusion. The International Journal of Robotics Research. 2023:02783649241273668.

---

### Official Review · Reviewer_Xdsd · 2024-11-04

**Soundness:** 3
**Presentation:** 2
**Contribution:** 3
**Rating:** 6
**Confidence:** 2

**Summary:**

This paper presents a method to use generative adversarial networks in an imitation learning framework. The general idea is to create a reward function using the concept of energy from a generative network which is a metric for how close a sample is to being from a distribution. They directly optimize on this energy metric as a reward and perform experiments in high dimensional locomotion settings. They find their method is able to outperform baselines in some situations but ends up struggling in situations with data scarcity.

**Strengths:**

The research problem for this work is good. Imitation learning is a reasonable method for many control tasks where reward is difficult to specify and expert data is available. Aiming to improve the imitation policy from samples is an interesting a relevant problem.

The novelty of this work seems good but it is slightly difficult to tell (see weaknesses).

The algorithm details seem good. It is unclear to me what “until horizon” means in line 230 and 231. Other than that the algorithm description is clear.

At a high level the experimental section is good. I think the presentation could be improved slightly and have questions about the baselines and statistical rigor. The discussion is interesting and provides insight into the method.

Ablation Studies are good. The discussion on the importance of annealing is interesting and relevant.

Failure Analysis is great. I really like the analysis and discussion of the reasons for failure. I think that leads to better future work and adds to the significance of this work.

Conclusion is good. I wish there was a future work section. Maybe more work into your annealing strategy?

**Weaknesses:**

I feel like as well the sentences from 71-77 are pretty vague and I would appreciate you explicitly stating the challenges clearly and then comparing your work even if is just “our method has smooth distributions and…”. The related work also seems to be wrapped in the intro which is fine but I would appreciate a more explicit comparison of other methods. Currently it simply says “this is what other methods do” instead of “here is how ours is different” which would make it easier to tell novelty.

The significance of this work seems good but it would be nice if the contribution was more explicitly stated. My interpretation was that the contribution is the use of energy based diffusion to train imitation learning but it would be better not to leave it up to the reader.

Baseline comparisons are ok (see questions).

The presentation of results is ok but I wish the tables were bigger sized, I’m pretty sure the page limit is 10 so you should be able to simply increase the size.

The statistical rigor is ok. You say each is trained 5 times and it is the results are averaged across 20 trials. 5 seems low to me here. Especially since the confidence intervals seem to be sort of wide and overlapping. Doing maybe 20 or 30 would be much better unless this is prohibitively expensive.

The clarity is ok. I didn’t find this paper very easy to read. Generally, I think more intuition could have been used in the paragraphs that starts at 167, 189 and 154. I’m admittedly not extremely versed in GANs but I think it should be written to provide more intuition to non GAN experts and more RL experts (as that is the target audience in my opinion). As well, I don’t feel the figures are helpful at all. I don’t understand what figure 1 is trying to convey nor do I understand even what it means. As well figure 3 also doesn’t make sense to me.

**Questions:**

iI there a reason you only compare to one baseline? Is that the only state of the art method and nothing can generally do better than it? I see line 364 says that it achieves superior results but is there truly no other baseline that even comes close? If so then that is fine.

Is this the only/first work that uses energy-based diffusion models? If it is I would appreciate you explicitly saying so.

---

> ### Author Response · Authors · 2024-11-16
> **Review responses & paper revisions**
>
> Dear reviewer, thank you for your feedback. We have revised the paper accordingly. Please find clarifications and paper changes below (text revisions based on your feedback are coloured blue). We hope these sufficiently establish the contribution and presentation of the paper.
>
> > It is unclear to me what “until horizon” means in line 230 and 231
> - Thank you for pointing this out. The horizon here refers to the rollout horizon used during policy training. The policy is rolled out for a horizon of 16 steps, then the collected transitions and rewards are added to the replay buffer to update the policy network weights. We have also explained this in detail in Appendix B.2.2 REINFORCEMENT LEARNING.
> - We realized the term “until horizon” might be unclear in the algorithm. To improve clarity, we have now explicitly mentioned this in the algorithm’s setup (line 219) and have pointed to the appendix for additional details.
>
> > I wish there was a future work section
> - Thank you for this suggestion. We agree that NEAR can be extended via several interesting ideas. We have now added a short future work paragraph on lines 498-511.
>
> > I feel like as well the sentences from 71-77 are pretty vague and I would appreciate you explicitly stating the challenges clearly and then comparing your work. The related work also seems to be wrapped in the intro which is fine but I would appreciate a more explicit comparison of other methods
> - We realize that lines 71-77 (in the old manuscript) could indeed be improved.
> - We have rewritten the introduction section (71 - 85) to clearly establish our contributions. We would appreciate it if the reviewer informs us if the revised one is clearer.
> - Related works relevant to NEAR rely on adversarial reward learning. This mechanism has been briefly explained in lines 62-71. We have rewritten lines 79-85 to clearly establish our contribution and differentiate it from the mechanism used in prior works. We would appreciate it if the reviewer could provide feedback on the updated lines.
>
> > The significance of this work seems good but it would be nice if the contribution was more explicitly stated. My interpretation was that the contribution is the use of energy-based diffusion to train imitation learning but it would be better not to leave it up to the reader.
> - Yes, the reviewer is correct in understanding that our contribution is to use energy-based models as reward functions to train imitation learning policies. Our approach allows us to learn this energy model offline and use it as a reward in the RL part in contrast with adversarial approaches that require an online update of the discriminator during the RL phase and lead to unstable min-max problems. We have modified lines 84-85 to state this explicitly.
>
> > I wish the tables were bigger sized, I’m pretty sure the page limit is 10 so you should be able to simply increase the size.
> - Yes, we agree that the results tables could be enlarged. In light of other reviews, we might have to add a few more details to the paper. For the moment we have not increased the table size but are happy to do so if there is space available after adding other suggested modifications. We hope the reviewer understands our prioritisation of clarifying content in the revised paper.
>
> > The statistical rigor is ok. You say each is trained 5 times and it is the results are averaged across 20 trials. 5 seems low to me here. Especially since the confidence intervals seem to be sort of wide and overlapping. Doing maybe 20 or 30 would be much better unless this is prohibitively expensive.
> - Yes, we understand that more trials would be beneficial. While running additional trials is not an unrealistic expectation, we chose to do 5 trials for the following reasons.
> - In our preliminary experiments on a subset of imitation tasks, we did not find a significant difference in results when using more trials.
> - While the computational requirements of a single trial are not too expensive, they grow to be significantly expensive for 20+ trials (across all 10 tasks in the paper). Unfortunately, being on a university-level computational budget, we are limited in terms of the available compute hours and GPU resources.
>
> > Generally, I think more intuition could have been used in the paragraphs that starts at 167, 189 and 154.
> - We have now included additional intuition behind score-based models and NCSN on lines 169, 182, 190. We hope this also sufficiently clarifies the idea of the DIST() function mentioned on line 208.

---

> > ### Author Response · Authors · 2024-11-16
> > **Review responses continuation**
> >
> > > As well, I don’t feel the figures are helpful at all. I don’t understand what figure 1 is trying to convey nor do I understand even what it means. As well figure 3 also doesn’t make sense to me.
> > - Thank you for pointing out the clarity issues with the figures. We feel that the figures are an important part of this paper and have made several changes to make sure that the reader understands their message. We have also revised the Figure 1 and Figure 3 captions to be more intuitive and clear to the reader. Additionally, we have now included another appendix section (B.3 MAZE DOMAIN DETAILS) that details the procedure to generate Figure 1. We have briefly explained both figures in the following lines and request the reviewer to provide feedback on the clarity of the revised figures.
> > - Figures 1 and 3 are a 2-dimensional running example that we use to illustrate the advantages of energy-based rewards (fig 1) and the annealing mechanism (fig 3). In this 2D domain (see fig 1 leftmost image), there is a red circular agent that aims to reach a yellow goal position (bottom right in the image). All expert demonstration trajectories (shown as a probability density $p_D$) pass through an L-shaped maze. It is expected that the learnt reward functions encourage the agent to pass through the maze.
> > - Figure 1 compares the reward functions learnt via energy-based and adversarial models. The energy based reward function is smooth and accurately captures the expert data density. In contrast, since the adversarial reward function is the output of a classifier, it is non-smooth and non-stationary. The adversarial reward's dependence on the agent's generated motions is shown by the low rewards in the distribution of policy-generated motions $p_G$.
> > - Figure 3 then explains the annealing mechanism using the same 2D domain. Here, again, $p_D$ is the expert's data distribution while $supp(\pi_{\theta_G})$ is the support of the distribution of states induced by rolling out policy $\pi_{\theta_G}$. Our energy-based model learns several perturbed versions of $p_D$. These are shown by the increasingly wide L-shaped boundaries (each L-shaped boundary is an energy function $e_{\theta}(., \sigma_{k})$). Between the two sub-figures in figure 3, the policy has improved from left-to-right (meaning $supp(\pi_{\theta_G})$ is closer to $p_D$). Annealing is shown by the change from $e_{\theta}(., \sigma_{k})$ to $e_{\theta}(., \sigma_{k+1})$. The annealed energy function provides better rewards to the agent.
> >
> > > Is there a reason you only compare to one baseline? Is that the only state of the art method and nothing can generally do better than it? I see line 364 says that it achieves superior results but is there truly no other baseline that even comes close? If so then that is fine. Is this the only/first work that uses energy-based diffusion models? If it is I would appreciate you explicitly saying so.
> > - To the best of our knowledge, AMP is the only comparable baseline for imitation learning from observation in such high-dimensional physics-based problems. It is an improved formulation of previous state-only methods like GAiFO [1] and is the strongest state-of-the-art baseline in this field (it is still a prominent algorithm used for learning such complex policies [5]). Below we briefly summarize the literature:
> >     - There are other score-based IL algorithms like [2] and its variants. However, these are not observation-based and require actions in their training data.
> >     - There are a few other algorithms that do state-only IL [3, 4]. However, none of these are directly comparable to either AMP or NEAR. For example CALM [3] can be thought of as a larger use case of the AMP idea and learns both high level (task level) and low level (motor level) policies by jointly training an ecoder and a conditional policy (it is mainly used for holistic character control). FAIL [4] learns policies via regret minimisation and decomposes the imitation problem into several independent min-max games (not comparable to the inverse RL formulation of NEAR).
> >     - Neither of these can be used to compare the effects of energy-based reward learning compared to adversarial reward learning.

---

> > > ### Author Response · Authors · 2024-11-16
> > > **References used in our response**
> > >
> > > [1] Torabi F, Warnell G, Stone P. Generative adversarial imitation from observation. arXiv preprint arXiv:1807.06158. 2018 Jul 17.
> > >
> > > [2] Chi C, Xu Z, Feng S, Cousineau E, Du Y, Burchfiel B, Tedrake R, Song S. Diffusion policy: Visuomotor policy learning via action diffusion. The International Journal of Robotics Research. 2023:02783649241273668.
> > >
> > > [3] Tessler C, Kasten Y, Guo Y, Mannor S, Chechik G, Peng XB. Calm: Conditional adversarial latent models for directable virtual characters. InACM SIGGRAPH 2023 Conference Proceedings 2023 Jul 23 (pp. 1-9).
> > >
> > > [4] Sun W, Vemula A, Boots B, Bagnell D. Provably efficient imitation learning from observation alone. InInternational conference on machine learning 2019 May 24 (pp. 6036-6045). PMLR.
> > >
> > > [5] L'Erario G, Hanover D, Romero A, Song Y, Nava G, Viceconte PM, Pucci D, Scaramuzza D. Learning to Walk and Fly with Adversarial Motion Priors. arXiv preprint arXiv:2309.12784. 2023 Sep 22.

---

> > > > ### Comment · Reviewer_Xdsd · 2024-11-25
> > > >
> > > > Thank you for your detailed response.
> > > >
> > > > Until horizon and future works is resolved for me.
> > > >
> > > > I appreciate the updated paragraph near like 75. It seems good enough to me and thank you for adding the contribution sentence.
> > > >
> > > > I see that you're now at the page limit. This means increasing the size of figures is difficult so it makes sense to leave it.
> > > >
> > > > I understand that you are limited on compute so it would be infeasible to add more experiments now. It would be nice if the paper gets accepted and you have more time to maybe double it to 10 just since 5 still seems extremely low and I'd imagine more trials would narrow the CIs at least a little bit.
> > > >
> > > > For Figure 1, I think I understand it but I think it still could be improved slightly. My initial interpretation was the reward function should be increasing as it gets closer to the goal (as that is generally how it will work) so the constant energy reward made it seem like the agent won't learn as the reward is simply high everywhere. I think the current caption may be benefited by saying something in text like "The energy-based reward is a smooth (each state has the same reward magnitude), accurate representation of...". Likely this is because I missed a definition of smooth in your text somewhere and I see that you have the "sum of rewards" notation in the caption but I would just like to have it super explicitly stated as this is a pretty important figure in my opinion.
> > > >
> > > > I'm still confused on Figure 3. So the idea is that you are decreasing the amount of noise in k+1, k+2... with the GAN right? Is that the idea? I thought it was more complex of a figure than that but if that is it then that makes sense. In the caption you say "the policy has improved from left to right" how is that known? The policy isn't being trained right the reward model is? Is that a typo?
> > > >
> > > > For baselines that is a reasonable response.

---

> > > > > ### Author Response · Authors · 2024-11-25
> > > > > **Thank you for the response (and additional clarifications)**
> > > > >
> > > > > Dear reviewer, thank you for the response. Here are some additional clarifications.
> > > > >
> > > > > > It would be nice if the paper gets accepted and you have more time to maybe double it to 10.
> > > > > - Thank you for pointing this out. We agree and will soon set up the additional trials. These at the very least will be added to the arXiv submission (and hopefully also to the main paper if obtained on time).
> > > > >
> > > > > > My initial interpretation was the reward function should be increasing as it gets closer to the goal (as that is generally how it will work) so the constant energy reward made it seem like the agent won't learn as the reward is simply high everywhere.
> > > > > - By constant energy reward, we mean to imply that the energy function is pre-trained and does not change during training. This is why in the updated Figure 1, the energy-based reward stays the same throughout training, while the adversarial reward changes through training time (we refer to this as non-stationarity in the adversarial reward).
> > > > > - To clarify your initial interpretation, yes the energy-based reward of course encourages the agent to get closer to the goal. We have now slightly modified the energy reward shown in Figure 1 (middle) to show that the reward conditioned on any state in the domain indeed encourages the agent to progress further in the maze.
> > > > >
> > > > > >  I think the current caption may be benefited by saying something in text like "The energy-based reward is a smooth (each state has the same reward magnitude), accurate representation of...". Likely this is because I missed a definition of smooth in your text somewhere ..
> > > > > - Thank you for pointing this out, we agree that Figure 1 is indeed quite important for the paper. By smooth, we mean that the reward function is continuous and provides informative gradients which are beneficial for reinforcement learning. In contrast, the adversarial reward is non-smooth, which means that its value (and gradients) can suddenly shift drastically. We believe that this non-smoothness in adversarial imitation learning would hinder RL.
> > > > > - We have now updated the caption to clarify what we mean by smooth. We have also updated the right subfigure in Figure 1 to better illustrate this smoothness/non-smoothness.
> > > > >
> > > > > > I'm still confused on Figure 3. So the idea is that you are decreasing the amount of noise in k+1, k+2... with the GAN right? Is that the idea? I thought it was more complex of a figure than that but if that is it then that makes sense. In the caption you say "the policy has improved from left to right" how is that known? The policy isn't being trained right the reward model is? Is that a typo?
> > > > > - Yes, you are correct to understand that the amount of noise decreases with $k$, $k+1$, $k+2$ ... (this is not related to the GAN and instead is the amount of noise added to the expert dataset when learning our score-based model). This essentially means that the energy function associated with $k+1$ is less dilated than the one associated with $k$ (as can be seen in the figure).
> > > > > - The idea of Figure 3 is to then illustrate annealing. Annealing is simply the process of changing the energy function (reward) from level $k$ to $k+1$ ... on based on the agent's improvement during RL. This isn't a typo and annealing instead happens during RL (given a set of pre-trained energy functions).
> > > > > - In Figure 3, the agent has improved from left to right since it is now generating motions that are closer to the expert distribution (notice that $supp(\pi_{\theta_G})$ is closer to $p_D$ on the right side). Given such an improvement, annealing means simply switching from the energy function (reward) of level $k$ to the one of level $k+1$.
> > > > > - We have now updated Figure 3 with additional headings and some changes to the caption.
> > > > >
> > > > > We hope this sufficiently clarifies the idea of our figures. We are happy to further clarify any questions and to make any suggested modifications.

---

### Meta-Review · Area_Chair_qxNb · 2024-12-21

**Metareview:**

The paper proposes an observation-only imitation learning which avoids the use of adversarial training for stability. The reviewers unanimously recommend weak acceptance. After the discussion, the main concern remains is the lack of baselines in the experiments. The paper only compares with AMP, but lacks simpler baselines like DeepMimic based on L2 distance which is also not based on adversarial training.

Some experimental results are also inconclusive (e.g. in Table NEAR and AMP have close win rates) potentially because the experimental results are based on 5 seeds only leaving high stdev. Despite the authors's arguments, I still think we need more baselines and seeds to establish the effectiveness of the proposed method. There're also recent work (like DIFO form "Diffusion Imitation from Observation" by Huang et al. 2024) which is also non-adversarial and fits into the setup here, though they show results on lower-dim problems. IL-flOW ("IL-flOw: Imitation Learning from Observation using Normalizing Flows" by Chang et al. 2022) proposed also a similar idea to this paper (learning a likelihood model on expert transitions and use it as reward for RL, which should also be discussed.

While I recommend acceptance at this point, I strongly encourage the authors to take reviewers' feedback, especially the points on baselines, expansion of related work discussion, adding more experimental results.

**Additional Comments On Reviewer Discussion:**

Reviewer Xdsd raises concerns on novelty, writing clarity, and choice of baseline. The concerns are addressed. Reviewer PK39 raises concerns on missing experiments on discrete and goal-oriented domains, training cost details, and choice of baseline. The main concerns are addressed. Reviewer Tb6J raise issues on writing clarity, choice of baseline, and unclear experimental results. Most concerns are addressed. Reviewer hjSn raises concerns on the lack of comparison with related work and novelty. The concerns are addressed.

In discussion, Reviewer Tb6J raises concern that Figure 1 might be misrepresenting the reward landscape of AMP, AMP actually uses the probability that a trajectory belongs to the dataset, rather than a 0/1 classification.

---

### Decision · Program_Chairs · 2025-01-22

Accept (Poster)